# Bacterial Microbiota of *Ostreobium*, the Coral-Isolated Chlorophyte Ectosymbiont, at Contrasted Salinities

**DOI:** 10.3390/microorganisms11051318

**Published:** 2023-05-17

**Authors:** Anaïs Massé, Juliette Detang, Charlotte Duval, Sébastien Duperron, Anthony C. Woo, Isabelle Domart-Coulon

**Affiliations:** 1Molécules de Communication et Adaptation des Microorganismes (MCAM), Muséum National d’Histoire Naturelle (MNHN), CNRS (UMR7245), CP54, 63 Rue Buffon, 75005 Paris, France; 2Pôle Analyse de Données UAR 2700 2AD, Muséum National d’Histoire Naturelle (MNHN), 43 Rue Cuvier, 75005 Paris, France

**Keywords:** Ulvophyceae, bacterial communities, microbiota, endosymbionts, salinity, 16S rDNA, metabarcoding, CARD-FISH

## Abstract

Microscopic filaments of the siphonous green algae *Ostreobium* (Ulvophyceae, Bryopsidales) colonize and dissolve the calcium carbonate skeletons of coral colonies in reefs of contrasted salinities. Here, we analyzed their bacterial community’s composition and plasticity in response to salinity. Multiple cultures of *Pocillopora* coral-isolated *Ostreobium* strains from two distinct *rbc*L lineages representative of IndoPacific environmental phylotypes were pre-acclimatized (>9 months) to three ecologically relevant reef salinities: 32.9, 35.1, and 40.2 psu. Bacterial phylotypes were visualized for the first time at filament scale by CARD-FISH in algal tissue sections, within siphons, at their surface or in their mucilage. *Ostreobium*-associated microbiota, characterized by bacterial 16S rDNA metabarcoding of cultured thalli and their corresponding supernatants, were structured by host genotype (*Ostreobium* strain lineage), with dominant Kiloniellaceae or Rhodospirillaceae (Alphaproteobacteria, Rhodospirillales) depending on *Ostreobium* lineage, and shifted Rhizobiales’ abundances in response to the salinity increase. A small core microbiota composed of seven ASVs (~1.5% of thalli ASVs, 19–36% cumulated proportions) was persistent across three salinities in both genotypes, with putative intracellular Amoebophilaceae and Rickettsiales_AB1, as well as Hyphomonadaceae and Rhodospirillaceae also detected within environmental (*Ostreobium*-colonized) *Pocillopora* coral skeletons. This novel knowledge on the taxonomic diversity of *Ostreobium* bacteria paves the way to functional interaction studies within the coral holobiont.

## 1. Introduction

Among reef benthic microorganisms, the siphonous green algae in the genus *Ostreobium* (Chlorophytes) are major carbonate bioerosion agents, dissolving up to a kilogram of reef carbonate per m^2^ per year [1]. Ubiquitous in intertropical, subtropical, and temperate reefs, they colonize carbonates from photic to mesophotic depths [2,3,4].

Forming an entire Ostreobineae suborder within the Bryopsidales order (Ulvophyceae class) [5,6], these genetically highly diverse algae are cryptic, growing exclusively as microscopic filaments. Their thallus is composed of a network of interwoven, branching, giant coenocytic filaments (siphons), which are 10 to 20 µm in diameter and up to several hundred µm in length, with multiple nuclei and chloroplasts that move by cytoplasmic streaming. When endolithic (bioeroding form), these filaments erode galleries within the carbonate, with epilithic three-dimensional “tufts of filaments” emerging at the substratum surface, which can be isolated as a free-living growth form [5,7,8]. In contrast, the coenocytic thalli of other siphonous Bryopsidale green algae, such as *Bryopsis* and *Codium* (Bryopsidineae) or *Caulerpa* and *Halimeda* (Halimedineae), form macroscopic three-dimensional algal bodies, with morphologically differentiated basal anchoring rhizoids and frond-like axes (sometimes bearing lateral branches and cortical utricles) [9].

Prevalent in the skeleton of reef-building corals, the Ostreobineae algal filaments are true endoliths, actively penetrating the hard carbonate via chemical means [1] and dominating the endolithic microbial communities of living colonies [10]. Coral species, which differ in their skeletal morphology and growth rate, can harbor different *Ostreobium* biomasses. In slow-growing massive corals, visible green bands can be formed by dense biomass beneath the host tissue [11,12]. In contrast, in fast-growing branched corals, networks of microboring filaments are less visible, “diluted” in the white skeleton mass just below the living coral tissues, and increasingly abundant towards branch bases [13,14]. Fordyce et al. [15] showed a correlation between skeletal density (which impacts light trapping) with biomass and chlorophyll concentration of phototroph endolithic communities dominated by *Ostreobium*. These eroding algae emerge in tissue-free areas, for example at the surface of fish bite scars, and after bleaching-induced coral decay and death [1,12,16]. Transmitted horizontally from free-living environmental propagules to the skeleton of juvenile coral recruits, these ectosymbionts were detected as early as 7 days after coral larval metamorphosis and the onset of calcification [14]. There is growing consensus that these photoautotroph algal endoliths and their associated prokaryotes have an overlooked functional role in the health of their coral host (reviewed by Pernice et al. [17], Ricci et al. [18], and van Oppen and Blackall [19]): they bloom during coral-bleaching episodes, and support bleached coral recovery via the suggested provision of alternative photosynthetic carbon nutrients [20,21,22]. Other suggested roles involve contribution to nitrogen cycling [7,23] and increased host photoprotection via modifications of skeletal optical properties [24].

Investigating the spatially resolved taxonomic diversity of endoliths within a single coral host colony is an emerging field. Multiple Ostreobineae genotypes co-exist with diverse prokaryote communities with a patchy spatial structure at the centimeter scale [25], which are typically more well-characterized and more genetically diversified in massive slow-growing than branched fast-growing coral host colonies, both in reef [26] and aquarium settings [14]. The localization and composition of the bacterial communities specifically associated with each *Ostreobium* genotype remains a black box, both for isolated strains and within natural coral host colonies. Visualization of bacteria in intact algal siphons (intracellular or at surface) is an open challenge. It remains to be determined whether distinct bacterial communities are associated with distinct *Ostreobium* genotypes and how these associations respond to abiotic factors.

In culture experiments, a pioneer study [7] detected fatty acid bacterial markers in *Ostreobium* strains isolated from aquarium-propagated *Pocillopora* coral colonies. A recent survey of the taxonomical composition of the bacterial communities associated with *Ostreobium* strains noted differences between phylotypes [27]; however, this was noted without robust comparison of multiple cultures for each strain and experimental manipulation of abiotic factors.

The salinity tolerance range of Ostreobineae algae and salinity’s impact on their associated bacterial communities is unknown. Reef-building corals and their associated algae thrive in a broad range of salinities in tropical and subtropical areas, from 20 psu in brackish estuaries to 40–42 psu in the Red Sea [28]. Salinity is globally affected by climate change, with centennial records showing seawater becoming saltier, for example in the Adriatic Sea (+0.18 psu in the last century [29]), and 1950–2000s records show wet areas getting wetter and having less saline, and dry regions getting saltier (global water cycle intensification [30]). Locally, in shallow water reef flats, salinity stress on the coral holobiont is caused by heavy rainfall and freshwater run-off during episodes of tropical storms and by evaporation during heat waves of increased frequency and severity. Hypersalinity stress is also caused by brine discharge from desalination plants in tropical locations adjacent to coral reef ecosystems [31]. High salinity is known to impact the physiology of tropical corals (which were aquarium-acclimatized to 34–40 psu salinities in Ferrier-Pages et al. [32] or reef-exposed to desalination plant brine in Petersen et al. [31]) but its effect on carbonate-associated skeletal microorganisms is unexplored. The bacterial communities of *Fungia granulosa* corals shifted composition after 29 days exposure to hypersalinity, with selection of bacterial taxa conferring benefits to the coral host [33], but this study did not focus on the skeletal endolithic compartment. Coral-isolated *Ostreobium* strains are in vitro models to experimentally investigate the microbiology of this alga in simplified systems [7,8] and could potentially reveal bacterial markers of algal adaptation to increased salinity.

Here, the diversity of *Ostreobium*-associated bacteria was investigated in two distinct algal host lineages, each exposed to three salinity levels. Experiments were conducted on genotyped strain representatives of *rbc*L clade P1 (010) and *rbc*L clade P14 (06) lineages sensu Massé et al. [7], which both belong to *Ostreobium* lineage 3 sensu Tandon et al. [34], within the Ostreobineae suborder. The culture medium salinity was manipulated in order to investigate *Ostreobium* tolerance to the salinity increase and related changes in its associated bacteria in multiple (*n* = 4) cultures of each strain, which were long-term acclimatized (9–13 months) to ecologically relevant low (32.9 psu), medium (35.1 psu), and high salinities (40.2 psu). CARD-FISH experiments on *Ostreobium* thalli acclimatized to low and high salinities (32.9 and 40.2 psu) allowed us to visualize at filament scale the bacterial phylotypes in algal tissue sections. Bacterial 16S rDNA gene amplicons’ sequencing characterized the taxonomic diversity of bacteria in the cultured *Ostreobium* thalli and their corresponding supernatants according to the genotype and salinity. Core bacteria, persistent across salinities and *Ostreobium* genotypes, were partly detected in the skeletal bacterial fraction of field-sampled, *Ostreobium*-colonized, coral endolithic communities, validating the ecological significance of these culture-based findings.

## 2. Materials and Methods

### 2.1. Ostreobium Strains and Acclimatation to Three Salinities

The bacterial communities associated with the two *Ostreobium* strains, 010 (MNHN-ALCP-2019-873.3) and 06 (MNHN-ALCP-2019-873.2) [7], from the RBCell ALCP collection of microalgae, Paris, France [35] were studied after long-term pre-acclimatization to three salinities: 32.9 psu (13 months), 35.1 psu (12 months), and 40.2 psu (9 months), which are representative of the natural salinity range in tropical coral reef ecosystems. Both strains were initially co-isolated in 2016 from a healthy branch tip of the same coral host colony *Pocillopora acuta* (Lamarck 1816) and propagated in long-term, closed-circuit cultures at the Aquarium Tropical, Palais de la Porte Dorée, Paris, France (initially imported from Indonesia before 2010). These *Ostreobium* strains have previously been genotyped to two distinct genetic lineages, i.e., clades P1 for 010 and P14 for 06, based on the *rbc*L taxonomic marker gene (>8% sequence divergence [7]). Free-living growth forms of each *Ostreobium* strain were cultured as suspended tufts of filaments in ventilated T-25 plastic flasks (Sarstedt, Nümbrecht, Germany) containing 25 mL of Provasoli Enriched Seawater medium (PES; [7,36]). *Ostreobium* strains were originally isolated from their coral host and propagated for 4.5 years with low doses of penicillin (100 U/mL) and streptomycin (100 µg/mL) antibiotics to prevent overgrowth by opportunistic bacteria [7]. Although routinely used in animal cell cultures, these generic antibiotics are known to be inefficient against most native marine bacteria [37]. Indeed, bacterial fatty acid markers were consistently detected within chloroform extracts of these strains [7]. However, to remove antibiotics’ selection pressure on the algal microbiota, penicillin was omitted for the last 9 months, and streptomycin was omitted for the last month before sampling for DNA extraction. A long-term ex situ culture introduces microbiota selection pressure but also reduces the variability of abiotic parameters, stabilizing algal phenotypes for identification of persistent core microbiota members and reproducible experimental testing of salinity response under in vitro controlled conditions.

For each *Ostreobium* strain and salinity levels, multiple cultures (*n* = 4) were used. For both strains, a parental culture stock (initial algal thallus) was split in two subcultures, each re-split in two after 6 to 8 months of growth, resulting in four distinct subcultures per strain (Appendix A). Initially grown at 29 psu, these subcultures were gradually acclimatized before the experiment to salinities 32.9 (low salinity, temperate NEAtlantic), then 35.1 (average salinity of Indo-Pacific seawater), and finally to 40.2 psu (Red Sea salinity) by gradual, 2–3 psu/month salinity increase, at each monthly medium renewal (Appendix A). Artificial seawater of each salinity was prepared by dissolving commercial Reef Crystals sea salt mix (Aquarium Systems, Sarrebourg, France) in H_2_O (38.6, 40, and 42 g/L for 33.5, 35.5, and 40.5 psu, respectively), adjusting pH to ~8.15 (average seawater pH), and then checking final salinity with a CDC401 (HACH) conductivity probe. Salinities of fresh and spent PES culture medium were controlled at monthly medium renewal, and the measured salinity drift was less than 5%. Cultures were incubated at 25 °C with 40 rpm orbital shaking (INFORS horizontally rotating platform incubator, Switzerland), under air, and a 12 h light/12 h dark cycle of illumination with white fluorescent light intensity of 31 ± 5.5 μmol.m^−2^.s^−1^ (measured with a spherical quantum sensor Li-Cor, Lincoln, NE, USA). Culture medium and T-25 flask were renewed every 4 weeks in all subcultures. Given their long-term (14–20 months), physically separate maintenance since sampling from initial stock cultures and before sampling for microbiota characterization, these multiple cultures were assumed to be independent biological replicates. During the year of acclimatization to the three studied salinities, all algal cultures were healthy, with growth and chlorophyll content conserved across the three salinity treatments, as measured with visible monthly biomass increase and stable green coloration of the algal tufts of filaments.

### 2.2. Catalyzed Reporter Deposition Fluorescence In Situ Hybridization (CARD-FISH)

*Ostreobium* thalli of two cultures of each strain (010, 06), acclimatized either to low or high salinities (32.9 and 40.2 psu), were scalpel microdissected into fragments (3 mm × 3 mm), allowed to recover for 1 week in vitro, fixed in 4% paraformaldehyde in Sorensen buffer (0.1 M) supplemented with sucrose (0.6 M) for 2 h at room temperature (RT), and then stored at 4 °C. Fixed algal thalli (*n* = 4) were rinsed in Sorensen buffer (0.1 M) and embedded in sterile 1.5 % agarose for protection of the fragile network of filaments, before dehydration in an increasing series of ethanol (70% to 100%). Ethanol was substituted overnight by butanol-1, and tissues were embedded in Paraplast Plus paraffin (Leica, France). Histological sections (10 µm) were cut with disposable blades on Leica RM2265 microtome and collected on Superfrost PlusTM slides. Sections were deparaffinized with 100% toluene, substituted in ethanol (100% then 96%), and air-dried. They were treated with 3% hydrogen peroxide solution (Sigma, St. Louis, MO, USA for 30 min at RT to inactivate endogenous peroxidases and then rinsed three times in 20 mM of Tris-HCl at pH 8.0. To allow for the oligonucleotide probe’s access to bacterial 16S rRNA, sections were treated with HCl (10 mM, 10 min at RT), lysozyme (10 mg/mL, 15 min at 37 °C), and proteinase K (1 µg/mL, 15 min at 37 °C) dissolved in 20 mM of Tris, with three rinses in 20 mM of Tris-HCl at pH 8.0 after each step. *Ostreobium* sections were hybridized using a EUB338mix of three probes: EUB338 (5′ GCTGCCTCCCGTAGGAGT-3) [38], EUB338 II (5′GCAGCCACCCGTAGGTGT 3′), and EUB338 III (5′ GCTGCCACCCGTAGGTGT 3′) [39] (*v*/*v*/*v* 2:1:1)—a combination that detects most bacteria. Negative controls were performed using non-EUB probe (5′ CTCCTACGGGAGGCAGC 3′) [38] on serial sections of the same paraffin block, mounted on the same slide. We checked that chloroplast-encoded 16S rRNA sequences (from published *Ostreobium* genomes) did not align (3 mismatches) with reverse sequences of EUB probes, excluding artefactual probe hybridization on chloroplasts. To amplify the hybridization signal, we used horseradish peroxidase-conjugated oligonucleotidic probes (HRP-probes, purchased from http://www.biomers.net/, Germany, consulted on 1 February 2022) diluted at 5 ng/µL in hybridization buffer (HB), composed of 0.9 M of NaCl, 20 mM of Tris-HCl at pH 8.0, 0.01% SDS, 10% dextran sulfate, 1% blocking reagent (Roche Diagnostics, Mannheim, Germany), and 35% (*v*/*v*) formamide (for both EUB338 mix and nonEUB). Each section was covered with ~10µL of HRP-probe in hybridization buffer (exact volume adjusted to cover the surface of each section). Hybridization was performed in wet slide chamber at 42 °C for 2 h 30 min, then stabilized at 43 °C for 15 min in washing buffer (5 mM of EDTA, 20 mM of Tris-HCl at pH 8.0, 0.05% SDS, 42 mM of NaCl, and 35% (*v*/*v*) formamide), followed by two steps of rinsing in 10 mM of PBS at pH 7.4 at RT. The HRP-probe signal was then amplified for 15 min at 37 °C in amplification buffer (PBS 0.8X at pH 7.4, 1.6 M of NaCl, 8% dextran sulfate, 0.08% blocking reagent) containing 10 µL of H_2_O_2_ 0.3% and 2.5 µL of Alexa488 fluorochrome-labeled tyramides (Invitrogen, Eugene, OR, USA). Sections were washed two times in PBS for 10 min at RT, rinsed for 1 min with milli-Q water, air-dried, and mounted in Fluoroshield containing DAPI (Sigma). Thalli sections (10 µm) were observed with a Zeiss AxioZoom V16 macroscope equipped with a X2.3 Plan NeoFluar objective (NA 0.57, FWD 10.6 mm), HXP200C metal halide lamp, and Zeiss AxioCam 506 camera. Fluorescence images were acquired in single plane or z-stack series (8–12 µm depth, step 1µm) and the same exposure settings for the treatment (EUBmix) and control (nonEUB) with ZenBlue 3.2 software at the MNHN photonic microscopy platform (CeMIM). Alexa488-positive bacterial morphotypes were visualized in extended depth of focus projection (Wavelets method) of a deconvoluted (nearest neighbor) z-stack series of images acquired at Ex 450–490 nm (Em 500–550 nm). Dapi-stained DNA was visualized at Ex 340–390 nm (Em 420–470 nm). Multichannel images were assembled in ImageJ (Fiji, open source software, https://fiji.sc) with similar visualization settings (intensity and contrast) for treatment (EUBmix) and controls (non EUB). Multiple algal tissue areas in 1–2 serial sections per treatment or controls were imaged per slide, with 1–2 slides imaged per culture.

### 2.3. Sampling and DNA Extraction

Biomass (thalli) of *Ostreobium* (*n* = 24, i.e., 4 biological replicates ×3 salinities ×2 strains, with dry weight 12.55 ± 4.54 mg) were sampled 9 days after transferring the cultures to fresh PES medium. Tufts of filaments were rinsed three times in ~20 mL of 0.22 µm filter-sterilized artificial seawater (Reef Crystal) at corresponding salinity levels and then lyophilized. The corresponding supernatants (*n* = 24) had been sampled previously at the last transfer (~25 mL each, after 1 month of *Ostreobium* culture), filtered through a 0.22 µm GTTP IsoporeTM membrane filter, and stored at −80 °C to access the bacterial community composition of strain supernatants. Total DNA extraction of algal biomass and supernatants (i.e., filters) was performed in sterile conditions (under laminar flow hood) using DNeasy PowerSoil™ Kit (Qiagen Laboratories, Hilden, Germany. For *Ostreobium* biomass, 2% polyvinylpyrrolidone (PVP) was added in the lysis buffer to remove phenolic compounds. For all DNA extracts, glycogen was added as a nucleic acid carrier (30 µg/mL final concentration) before DNA precipitation. As internal controls, DNA extractions were also performed on fresh PES medium for each salinity (in duplicate, *n* = 6, i.e., 3 salinities ×2), 2% PVP (*n* = 1), and a membrane filter (*n* = 1).

### 2.4. PCR Amplification and Illumina Sequencing

The V5–V7 region of the bacterial 16S rDNA (~400 bp) was amplified using the specific primer pair 799F (5′ AACMGGATTAGATACCCKG 3′) and 1193R (5′ ACGTCATCCCCACCTTCC 3′) to limit co-amplification of chloroplast 16S rDNA of the algal host [40,41]. Illumina adapter sequences were added to the 5′-end of the forward and reverse primers. To increase the recovered diversity, PCR amplifications were performed in triplicate for each sample, then pooled during the purification step (see below). Amplification reactions were performed in 25 µL volume containing 2 µL of DNA extract template, 0.5 µM of each primer, 2 µL of MgCl_2_ (25 mM), 0.5 µL of DNTPmix (10 mM), 5 µL of Go Taq Flexi Buffer, and 0.125 µL of 5X GoTaq^®^ G2 Flexi DNA Polymerase (Promega, Madison, WI, USA) in sterile water. The PCR cycling conditions were as follows: 3 min at 95 °C; 35 cycles of 20 s at 95 °C, 30 s at 54 °C, and 30 s at 72 °C); and 10 min terminal extension at 72 °C. Amplified fragments were visualized in 1% agarose gels with SYBRSafe, and amplicons from 3 independent positive PCR reactions were pooled and then column purified (Nucleospin^®^ gel and PCR clean-up kit, Macherey-Nagel, Hœrdt, France) in EB elution buffer (Qiagen). Amplicons from negative PCR controls were also pooled and purified as an internal control (*n* = 1). All purified amplicons (25 µL) were submitted for MiSeq sequencing (Illumina, paired-end, 2 × 300 bp) to Eurofins Genomics (Ebersberg, Germany).

### 2.5. Sequence Dataset Analysis

Primer sequences were removed in raw reads using CUTADAPT program (v2.5, [42]). Paired 16S rDNA amplicon reads were then assembled using FLASH algorithm (2.2.00, [43]) with a minimum overlap size of 10 bp. Sequences analysis was performed using QIIME 2 2022.2 [44]. Consensus sequences were demultiplexed using DEMUX algorithm. The DADA2 plugin was used to remove chimeras, trim sequences to 368 bp, denoise the dataset, and assign reads to Amplicon Sequence Variants (ASVs). ASVs were assigned to taxonomic groups using the feature-classifier plugin and classify-sklearn module in SILVA v138 SSU rRNA database (released 16 December 2019). ASVs affiliated with eukaryota, chloroplast, mitochondria, and unassigned ASVs (i.e., assigned to none of the three domains of life) were removed from the dataset, as well as all ASVs of the 9 internal controls (ASVs were detected either exclusively in fresh PES medium at each salinity, PVP extraction supplement, membrane filters, and PCR controls, or shared with at least one algal (thalli or supernatants) sample, without considering relative abundances (thresholds). This stringent approach (manually removing all ASVs shared with blanks) may underestimate natural bacterial diversity but was compensated by the robust within-strain comparison of multiple cultures (12 per each strain). In *Ostreobium* biomass samples, bacterial sequences unassigned below the class level in SILVA v138 were then aligned by BLASTn with 16S RNA-encoding sequences available in GenBank database (consulted on 21–22 July 2022): the order could be identified when the unassigned sequence had a similarity of more than 90% with the Genbank reference sequence. Rarefaction curves were performed on rarefied ASV table using QIIME 2 2022.2 (1080 sequences (minimum library size) sub-sampled for each sample). 

Alpha diversity (ASV richness, Shannon and Pielou’s evenness indices) was analyzed on the unrarefied dataset in R software v4.0.5 (package vegan). Differences between groups (salinity treatment and *Ostreobium* genotypes) were tested using the non-parametric Kruskal–Wallis test, followed by a pairwise post hoc analysis of Mann and Whitney with a threshold value of 5% (α = 0.05). For beta diversity, a matrix of Bray–Curtis distances was calculated from the arcsine square root-transformed unrarefied bacterial ASV proportions. Bacterial communities were compared between algal categories (*Ostreobium* thalli vs. supernatants), genetic lineages (strain 010 clade P1 vs. strain 06 clade P14), salinities (32.9 vs. 35.1 vs. 40.2 psu), and their interactions using permutational multivariate analysis of variance implemented in the ‘adonis2′ function of the vegan package in R (*n* = 999 permutations, α  =  0.05) followed by pairwise comparisons (‘pairwise.adonis’ function) with Bonferroni correction.

To plot the bacterial variation among sample categories (thalli and supernatant for each strain), Principal Coordinates Analysis (PCoA) of Bray–Curtis distances was performed in R (v4.0.5) on bacterial ASV proportions (arcsine square root-transformed unrarefied data). Another PCoA focused on *Ostreobium* thalli to visualize the microbiota variance structure among algal genetic lineages and salinities.. Then, Sparse Partial Least Square Discriminant Analysis (sPLS-DA) modeling was used to classify by host lineage or salinity the bacterial community profiles (R v3.6.2, package mixOmics, arcsine square root-transformed unrarefied proportions) with analysis limited to the 1000 most discriminant ASVs of the high-dimension dataset. Next, to select informative ASV variables, that predict community projection on the principal component dimension that best discriminates between factors, we further analyzed the correlation of sample projection coordinates with ASV composition. We thus ranked bacterial ASVs with correlation coefficient > 0.7 towards the sample projection on Component 1 of the PLS models. This allowed for the identification of bacterial ASVs that contribute the most to the classification of groups, and highlighted the most discriminating bacterial ASVs within each model. Subsequent heatmaps illustrated fold changes of these differentially expressed variables among replicates and treatments (R v4.0.5, package “pheatmap”). Data were visualized with R v4.0.5 packages “ggplot2” and “tidyverse”. Venn diagrams were also built to identify and visualize the ASVs that are part of *Ostreobium* core microbiota, the criterion being that these ASVs should be shared by both genetic lineages and occur across all 3 salinities. 

### 2.6. Data Availability

Sequences generated via Illumina Miseq sequencing during this study were deposited as NCBI project archive “BioProject: PRJNA896951”. Raw bacterial ASV tables (read counts of unfiltered and filtered data), fasta sequences, and corresponding taxonomy were deposited at figshare with doi: 10.6084/m9.figshare.21953075 (Appendix A).

## 3. Results

### 3.1. In Situ Localization of Bacteria Associated to Cultured Ostreobium Siphons

Bacterial phylotypes, visualized by CARD-FISH experiments using universal bacterial probe EUBmix (EUB338 + EUBII + EUB III) hybridized to 16S rRNA, were localized within or near *Ostreobium* filaments for strains of both genetic lineages and at low (32.9 psu) and high (40.2 psu) salinities (Figure 1A–D). Reconstructions in 3D of z-stacked series of observations showed an EUB-positive signal at the surface of *Ostreobium* siphons, indicative of epiphytic bacteria. An EUB-positive signal was also recorded within the siphons, revealing endophytic bacteria (with some uncertainty due to epifluorescence imaging corrected by signal deconvolution treatment). Indeed, controls using the non-EUB probe were negative (Figure 1A*–D*), and the positive signal was distinct from chloroplast structures. Positive CARD-FISH signals also revealed bacteria inside the mucilage between *Ostreobium* siphons (Figure 1). Based on the hybridization signal intensity, which relates to ribosomal RNA content, the detected bacteria were metabolically active. Typical of the algal siphoneous cytological organization, the nuclei counterstained with DAPI were heterogeneously distributed along the *Ostreobium* filaments (Figure 1), with empty portions alternating with areas of several elongated nuclei, concentrated near branching nodes, or spread along the siphon, together with discoidal chloroplasts. Chlorophyll autofluorescence was low here due to ethanol depigmentation before paraffin embedding (but observed in native filaments). Bacterial abundances were variable depending on algal structures, locally concentrated at the surface of siphon envelopes and in the mucilage, and seemed especially abundant at high salinity. These qualitative presence/absence results await quantitative confirmation to test putative differential distribution pattern between structures and cultures.

### 3.2. Quality of Illumina Sequencing Targeting Bacterial 16S rRNA V5–V7 Gene Fragment

A total of 7,099,477 sequences of the V5–V7 region of 16S rDNA were obtained from the 57 samples (*n* = 24 algal thalli, *n* = 24 supernatants, *n* = 9 internal controls) using Illumina MiSeq sequencing. After chimera detection and quality filtering, between 52 and 89.8% of the sequences were retained from each sample. These represented a total of 1564 distinct bacterial ASVs (unrarefied dataset) after removal of ASVs affiliated with eukaryota (2 ASVs) and unassigned sequences (4 ASVs) (0 AVS affiliated to chloroplast or mitochondria detected in these samples) and removal of internal controls (all 1617 ASVs detected, no matter their relative abundances—including 138 ASVs shared between controls and at least one sample of algal thalli or supernatant). Rarefaction curves of *Ostreobium* thalli and their corresponding supernatants (Appendix A, rarefied ASV table) reached a plateau indicating that a reasonable sequencing depth had been attained. 

### 3.3. Diversity and Structure of Bacterial Communities in Cultured Ostreobium

Alpha-Diversity indices measured at high taxonomic resolution (ASVs) on unrarefied data for *Ostreobium* algal thalli and their corresponding culture supernatants are presented in Figure 2 as a function of genotype and salinity (individual values of ASV richness and Pielou’s evenness indices are provided in Appendix A). Between 24 and 93 bacterial ASVs were detected in each individual sample. Cumulated ASVs, specific or shared between *Ostreobium* thalli and their supernatants, are illustrated in Venn diagrams (Figure 3A). Bacterial ASVs conserved in thalli across salinities are illustrated in Figure 3B,C for each strain (between strains comparison of thalli (Figure 3D) define the core microbiota, see Section 3.4).

In algal thalli, a total of 837 cumulated ASVs were detected from both *Ostreobium* genotypes at all three salinities (total *n* = 24 cultures; Figure 3A) with totals of 469 and 400 ASVs in the biomass of strains 06 (*n* = 12; Figure 3B) and 010 (*n* = 12; Figure 3C), respectively (similar average 54 ± 16 ASVs per sample for 06, and 49 ± 15 ASVs for 010; Figure 2A and Appendix A). A total of 809 cumulated bacterial ASVs were detected in supernatants from both genotypes at all three salinities (*n* = 24, Figure 3A). Between 9.2% and 13% of bacterial ASVs were shared between *Ostreobium* thalli and their corresponding supernatants for strains 06 and 010, respectively (Figure 3A). However, at a given salinity, only 5.7 to 8.8% of ASVs were shared between thalli of a genotype and its corresponding supernatants. 

Alpha-diversity equitability (Pielou’s evenness) in cultured *Ostreobium* algal thalli of strain 010 was stable across salinities and higher than in strain 06, with average values of 0.79 ± 0.03 (*n* = 4), 0.74 ± 0.10 (*n* = 4), and 0.76 ± 0.07 (*n* = 4) for salinities of 32.9, 35.1, and 40.2 psu, respectively (Figure 2B). In *Ostreobium* thalli of strain 06, Pielou’s evenness was lower but increased with salinity from 0.47 ± 0.15 (*n* = 4) to 0.51 ± 0.07 (*n* = 4) and then to 0.60 ± 0.11 (*n* = 4) at 32.9, 35.1, and 40.2 psu, respectively. Among thalli, Pielou’s evenness index was significantly different between genetic lineages (strains) at 32.9 and 35.1 psu (Figure 2B; *p* = 0.029), but not at 40.2 psu (*p* = 0.11). Overall, these results indicate high differentiation and even distribution of microbiota across salinities for thalli of strain 010. For thalli of strain 06, a pattern towards diversification (increased Shannon index, Appendix A) and homogenization (higher Pielou’s evenness index, Appendix A) was detected at highest salinity (40.2 psu). Among culture supernatants, Pielou’s evenness was stable whatever the salinity, with average values of 0.70 ± 0.12 (*n* = 12) for 010 and 0.71 ± 0.08 (*n* = 12) for 06.

The structure of bacterial communities associated to cultured *Ostreobium* thalli and their corresponding supernatants was visualized by Principal Coordinate Analysis (PCoA) of Bray–Curtis distances measured at the ASV level for strains 06 and 010 separately (Figure 4A). Whatever the strain (genotype) or the salinity, the microbiota profiles of cultured *Ostreobium* thalli overlapped with corresponding supernatants when projected on Component 1 (14.6%, PC1 for 06 and 12.2%, PC1 for 010) and Component 2 (10.5%, PC2 for 06 and 13.9%, PC2 for 010) (Figure 4A). Within cultured *Ostreobium* thalli (Figure 4B), Principal Coordinate Analysis of Bray–Curtis distances showed separate projections of bacterial communities according to algal genotype on Principal Component 1 (Figure 4B, 21.6%, PC1) and some structuration by salinity on Principal Component 2 (especially for strain 010, Figure 4B, 10.1%, PC2; also visible for strain 06 in Figure 4A, 10.5%, PC2).

Statistically (pairwise adonis comparisons), there was no significant difference between communities of supernatants and corresponding *Ostreobium* thalli (F = 0.973, df1, *p* = 0.467; Appendix A), but differences were significant between thalli genotypes (strain 06: *rbc*L clade P14 vs. strain 010: *rbc*L clade P1, F = 3.049, df1, *p* = 0.001), between salinities (*p* = 0.001, F values reported in Appendix A), and between the interaction of algal genotype and salinity (F = 1.889, df1, *p* = 0.002). Each genotype had a specific bacterial profile that responded differently to salinity increase. 

### 3.4. Taxonomic Composition: Identification of a Core Ostreobium Bacterial Microbiota

The bar plot distribution of bacteria relative abundances in *Ostreobium* thalli and their supernatants is illustrated at order level in Figure 5 (also at class level, Appendix A; taxonomy from SILVA v138 16 December 2019). Both Rhodospirillales and Kiloniellales orders have since been reclassified as distinct family-level lineages (Rhodospirillaceae and Kiloniellaceae) in the class Rhodospirillales (NCBI taxonomy browser, [45]).

A total of 54 bacterial orders (17 classes) were detected, of which 30 (6 classes) had relative abundances below 1%, with high interindividual variability (Figure 5; also at class level: Appendix A) among genotypes and salinities. The most abundant class detected across all samples was Alphaproteobacteria (15.6–94.2% cumulated abundances), represented by 20 orders (8 < 1%). Within this class, the most abundant and prevalent order in thalli of strain 010 (clade P1) was the Kiloniellales (33.3 ± 16%, *n* = 12). This order was also detected in all corresponding supernatants with similar relative abundances of 31 ± 18.9% (*n* = 12). In contrast, in thalli of strain 06 (clade P14), the Rhodospirillales was the most abundant (36.8 ± 19.9%, *n* = 12) and prevalent (100%) alphaproteobacterial order; however, it was much less abundant in the corresponding supernatants (2.9 ± 2.3%, 75% prevalence, *n* = 12). 

The Bacteroidia class was also detected in all samples, but at much higher abundances in supernatants (21 ± 19.4%, *n* = 24) than thalli (7.0 ± 5.6%, *n* = 24). In contrast, the Gammaproteobacteria, Acidimicrobiia, and Phycisphaerae classes were more abundant in thalli than the corresponding supernatants. Regarding the Gammaproteobacteria, this class was detected in all thalli with highly variable abundances (0.35 to 65.6%, average of 9.0 ± 16.5%, *n* = 24) and also detected in 21/24 supernatants but at 10-fold lower relative abundances (0.05 to 5.9%, average of 0.81 ± 1.28%, *n* = 24). Gammaproteobacteria were represented by 26 bacterial orders (20 < 1%) and dominated by Alteromonadales and Cellvibrionales, which were detected in 4/12 (0.95 ± 2.15%) and 11/12 (3.7 ± 2.9%) thalli of strain 010, and in 6/12 (11.3 ± 23.3%) and 12/12 (1.6 ± 2.9%) thalli of strain 06, respectively, and they were also detected at much lower abundances in all corresponding supernatants (*n* = 12, average < 0.9%). Regarding Acidimicrobiia (Microtrichales order) and Phycisphaerae (Phycisphaerales order), they were detected in 22/24 and 18/24 of *Ostreobium* thalli, respectively, at variable relative abundances (0.8–25.6% and 0.03–20%, respectively). They were less prevalent and abundant in supernatants (19/24 and 0.05–4.8%; 13/24 and 0.1–10.9%, respectively). 

Specific to thalli of strain 010, the Thermoleophilia class (Gaiellales order) was detected in 11/12 samples (3.1 ± 1.9%, *n* = 12) compared to none in strain 06; it was also detected at a trace level in strain 010 supernatants (5/12, 0.13 ± 0.20%). Finally, the Deltaproteobacteria class (former Polyangia, Myxococcia, Leptospirae, and Desulfovibrionia classes, cf NCBI taxonomy browser) was represented by only the Nannocystales order, detected at high salinity (40.2 psu) only in thalli and supernatants of strain 010 (3/12 and 3/12 with relative abundances of 1.2 ± 2.5% and 0.4 ± 1.1%, respectively). 

Venn diagrams revealed the strain-specific core microbiota, i.e., the core bacterial fraction persistent in thalli across salinities and specific to each genotype (Figure 3B,C), and the *Ostreobium* core microbiota, i.e., the core bacterial fraction shared by both genotypes and all three salinities (Figure 3D). Bubble plots summarize the proportion of each core bacterial order/ASV per category (Figure 6), with a detailed classification of shared ASVs among algal thalli and supernatants in Appendix A. Additional information is provided (Figure 6, detailed in Appendix A) on bacterial ASVs and orders detected in naturally occurring *Ostreobium*–bacteria assemblages within *Pocillopora* coral skeletons for environmental contextualization of the findings obtained in culture-based experiments.

Among the ASVs persistent across three salinities, only seven core bacterial ASVs were shared between thalli of both genotypes (1.8% out of a total of 400 bacterial ASVs for strain 010, 18.6% cumulated abundance; 1.5% out of a total of 469 bacterial ASVs for strain 06, 35.7% cumulated abundance; Figure 3D and Appendix A). This core *Ostreobium* microbiota was dominated by Alphaproteobacteria (5 ASVs), with one additional ASV belonging to Bacteroidia and another to Acidimicrobiia. 

Within the Alphaproteobacteria, the numerically dominant and most prevalent taxon in *Ostreobium* core microbiota was ASV16 (unresolved at genus level, classified to the family Rhodospirillaceae, order Rhodospirillales), with 75–100% prevalence, and it was 10 times more abundant in thalli of 06 (~32 %) than 010 (~3%) (Figure 6 and Appendix A). Additionally present in supernatants (<6.3% in 06 and <1% in 010), ASV16 was detected at trace level in coral skeletons (1/8; <0.01% abundance; Figure 6 and Appendix A). 

The second dominant core alphaproteobacterial taxon was ASV57 (family Hyphomonadaceae, order Caulobacterales), with 75–83% prevalence and 1.8 to 5.7% average relative abundances in 010 and 06, respectively (Figure 6 and Appendix A). It was detected at lower prevalence and relative abundances in supernatants and at trace levels in coral skeletons.

Three additional core alphaproteobacterial taxa were detected (<1% in 06 and 1.7–3.2% relative abundances in 010; Figure 6 and Appendix A). The Rhizobiales order (reclassified as Hyphomicrobiales) was represented by ASV72 (genus *Labrenzia*) and was highly prevalent (91%) in both strains. It was also detected in supernatants (at lower prevalence and abundances) and undetected in coral skeletons. The Kiloniellales order (reclassified as Kiloniellaceae family) was represented by ASV124 (genus *Fodinicurvata*) and detected in both strains at all salinities (33–66% prevalence). It was also present in supernatants and undetected in coral skeletons. Finally, the Rickettsiales order was represented by ASV76 (basal lineage (AB1) defined via the metagenome assembly of rare bacterial taxa in the marine bryozoan *Bugula neritina* [46]) and detected at low abundances (<2%) but with high to medium prevalence in thalli of both strains (100% in 010 and 50% in 06). It was also detected with the same prevalence but higher abundances in supernatants and at trace levels in coral skeletons (25% prevalence).

Within the Bacteroidia class, ASV166 (genus *Candidatus Amoebophilus*, family Amoebophilaceae, order Cytophagales) was a core bacterium with 100% prevalence and low abundances (<2%) in thalli of both strains. It was very rare in supernatants (2/24, average abundances <0.02%), but persistently detected (75% prevalence, at trace levels) in the skeletons of environmental *Pocillopora* corals.

Finally, within the Acidimicrobiia class, ASV167 (genus *Ilumatobacter*, order Microtrichales) was a core bacterium persistent across salinities in both strains (50% prevalent in 06, 75% prevalent in 010) at low abundances (0.27% in 06 and 1.86% in 010).

Additionally, in each strain we detected a strain-specific core microbiota (additional to the 7 ASVs presented above). For strain 010, it was composed of 12 ASVs (3.0% of a total of 400 bacterial ASVs), and for strain 06, it was composed of 19 ASVs (4.1 % of a total of 469 bacterial ASVs) (Figure 3B and Appendix A). Several taxa were represented by multiple ASVs, with cumulated abundances varying between algal genotypes (Figure 6). Mostly, within Kiloniellales, cumulated abundances of *Fodinicurvata* and *Pelagibius* reached 18.9% in strain 010 vs. only 1.3% in 06. Within Rhodospirillales, the Rhodospirillaceae family amounted to 5.8% in 010 vs. 34.8% in 06. Within Rhizobiales, the combined *Labrenzia*, unresolved Ahrensiaceae family members, and *Devosia* were more abundant in 010 (8.3%) than in 06 (1.8%). Within Microtrichales, *Illumatobacter* cumulated abundances were only 1.9% in 010 vs. 5.5% in 06. Finally, additional families were represented in only in one strain. Indeed, unresolved Rhodobacteraceae and *Paracoccus* were detected in 06 at 11.7% (<0.1% in 010). Distinct Cyclobacteriaceae were detected in 010 and in 06. *Muricauda* Flavobacteriaceae, Phycisphaeraceae SM1A02 lineage, and the Gaiellales (unresolved family) were detected only in 010.

### 3.5. Genotype-Driven Structuration of Ostreobium Microbiota, with Limited Salinity Effect

Sparse Partial Least Squares Discriminant Analysis (sPLS-DA) of bacterial ASV relative abundances confirmed the classification of bacterial profiles by the algal host genotype or abiotic salinity factor. Informative ASVs explaining microbiota differentiation were highlighted via correlation analysis of sample projection coordinates with composition.

For the factor genotype, sPLS-DA results showed the projection of bacterial communities into two clearly separated groups (Figure 7A, 6.57%, PC1), demonstrating that microbiota variability was structured by strain genotype. All four of the most discriminant bacterial ASVs (4 selected ASVs with correlation coefficient >0.7 for projection on PC1, Figure 7B) belonged to the core microbiota and were differentially expressed between 06 and 010. Indeed, the Rhodospirillaceae ASV16 was significantly more abundant in 06 than 010. Alternately, the three core ASVs were more abundant in 010 than 06: ASV176 (Kiloniellaceae), ASV171 (Cyclobacteriaceae), and ASV200 (Gaiellales).

For the factor salinity (analyzed separately for 06, Figure 8A; and 010, Figure 8B), sPLS-DA results showed that the microbiota of each strain (genotype) responded differently. For strain 06 (Figure 8A), the bacterial communities’ projections on PC1 (11.94% explained variability) were clearly separated at high salinity (40.2 psu) from the overlapped intermediate (35.1 psu) and low (32.9 psu) salinities. For strain 010 (Figure 8B), it was the bacterial communities at low (32.9 psu) salinity that were clearly projected separately on PC1 (13.47% explained variability) from the overlapped intermediate (35.1 psu) and high (40.2 psu) salinities. Interestingly, a single salinity-responsive, most discriminant ASV597 affiliated to the genus *Devosia* (Rhizobiales) was shared by thalli of both strains 06 and 010 (more abundant at high salinity). Additionally, in 010 (Figure 8C), four other ASVs were significantly responsive to salinity (more abundant at high salinity), including ASV57 (a Caulobacterales Hyphomonadaceae from the core microbiota) in addition to ASV170 (Nannocystales, Deltaproteobacteria), ASV383 (unclassified), and ASV976 (Kiloniellales). 

## 4. Discussion

The microbial phycosphere of bioeroding *Ostreobium* algae (Ulvophyceae) and its response to changing abiotic factors should be explored given the ecological importance of these cryptic colonizers of marine carbonates and reef corals. This study is the first to visualize bacterial phylotypes within/at the surface of *Ostreobium* siphons and to characterize the sensitivity of its bacterial communities in response to salinity. We also identified core *Ostreobium* bacteria, including potential intracellular symbionts, that are persistent across salinities and algal genotypes.

### 4.1. Small-Sized Core Ostreobium Bacterial Microbiota, Structured by Algal Genotype

Our experimental approach used multiple cultures (*n* = 12) of two *Ostreobium* strains from genetically distinct lineages [7], to reveal persistent core bacteria—shared vs. specific to each strain—amongst the highly variable bacterial microbiota observed between replicate cultures of the same strain (4 per each of 3 salinity levels). This large interculture variability is consistent with that reported in cultures of the green microalga *Tetraselmis suecica* strain F&M-M33 [47], where opportunistic bacteria may be enriched or introduced and carried over during successive subcultures separated from initial inoculum. A similar approach has been used to identify core bacteria in multiple cultures of long-term propagated strains of Symbiodiniaceae microalgal endosymbionts of coral tissues [48] in response to temperature [49], along with FISH visualization in hospite [50]. 

Here, the core bacterial taxa shared by thalli of both *Ostreobium* genotypes across all three salinities corresponded to a small subset of seven ASVs (1.5–1.8% of the detected ASVs in each strain), classified to seven distinct families. This result agrees with the small-sized core microbiota of another siphonous Bryopsidale, *Caulerpa* sp., where <1% of the total detected taxa were shared across four *Caulerpa* species [51].

*Ostreobium* bacterial communities were dominated by Rhodospirillales, either Rhodospirillaceae or Fodinicurvataceae (and other Kiloniellaceae) with differential abundances depending on host genotype. 

Rhodospirillaceae ASV16 (100% sequence identity with JQ516442.1 of a bacterial clone from the Caribbean coral *Orbicella (Montastrea) faveolata*; [52]) were abundant in algal thalli (especially of strain 06). Multiple additional Rhodospirillaceae were also detected (44 ASVs in 06, and 39 ASVs in 010), congruent with the reported dominance of two Rhodospirillaceae species (~55–60% sequence relative abundances) in the microbiome as putative endophyte of the Gulf of Mexico siphonous green alga *Caulerpa ashmeadii* [53]. Fodinicurvataceae (ASV124 and multiple other ASVs) were also abundant in algal thalli (especially of strain 010) and have been involved in overall N metabolism in another Bryopsidale alga, *Caulerpa prolifera* [51]. Kiloniellaceae have also been documented recently in a few *Ostreobium* strains [27] and within skeleton assemblages of several coral species [54]. Fodinicurvataceae are halophilic bacteria, abundant in intertidal sandflat algal mat environments [55], with type strains of the *Fodinicurvata* genus isolated from salt mine sediments [56]. Of note, the nitrate-reducing strain *Fodinicurvata* halophila isolated from a marine saltern contains C18:1ω7c as a major fatty acid (36%; [57]), which was previously detected in both studied *Ostreobium* strains [7]. Based on function prediction hypotheses inferred from the metabolism of cultured type strains, a functional category related to nitrogen metabolism was thus a dominant feature of the *Ostreobium* core microbiome, including not only the Fodinicurvataceae, but also two other alphaproteobacterial taxa, the Hyphomonadaceae (Caulobacterales) and the Stappiaceae (Rhizobiales). Nitrate-reducing, strictly aerobic bacteria in the family Hyphomonadaceae (here represented by ASV57) are adapted to oligotrophic environments in the ocean [58] and were already detected in marine macroalgae, including the kelp *Nereocystis luetkeana* [59], the seagrass *Halophila stipulacea* [60], and the red alga *Porphyra yezoensis* [61]. Within Hyphomonadaceae, the species *Algimonas porphyrae* isolated from red alga *Porphyra yezoensis* is also characterized by high content of the fatty acid C18:1ω7c [61], which is abundant in the studied *Ostreobium* strains [7].

Within the Stappiaceae, the genus *Labrenzia* (ASV72) was identified here as a core *Ostreobium* bacteria, in agreement with previous occurrence reports in five *Bryopsis* species (Bryopsidales) from long-term unialgal laboratory cultures [62] or sampled across Atlantic, Mediterranean, and Pacific localities, at salinities ranging from 32.4 to 37.6 psu [63]. *Labrenzia* bacteria have also been detected in the core microbiota of cultured Symbiodiniaceae dinoflagellates, which are major endosymbionts of coral tissues [48]. Members of Rhizobiales are known to fix dinitrogen (N_2_) into dissolved ammonium and nitrate, which are transferred to their plant host and promote its growth [64]; thus, these could play an important role in the nutrition and growth of *Ostreobium*, both in vitro and within reef environments. Indeed, carbonate-bioeroding *Ostreobium* form an important component of the benthic turf algae compartment, which is known to fix nitrogen [65]. Bioeroding *Ostreobium* typically colonize nitrogen-limited environments, such as biomineralized CaCO_3_ coral skeletons (with <1% dry weight of organic matter), and their epilithic filaments and free-living propagules are adapted to N-limited oligotrophic reef seawater. Overall, these *Ostreobium* core bacteria should be further studied for their potential significance in nitrogen assimilation and cycling.

Amongst other core *Ostreobium* bacteria, the genus *Ilumatobacter* (order Microtrichales) was represented in thalli of strain 06 by three ASVs and in strain 010 by two ASVs, with the minor detection in supernatants suggesting tight association. This genus was first described from bacteria isolated from beach sand off the Sea of Japan and is also associated with marine sponges [66]. The genome of the sponge-isolated cultured strain contains biosynthetic enzymes for bacterial vitamins (B6, K), omega-3 fatty acids, and xanthin carotenoids, and it lacks several amino acid biosynthetic enzymes, but encodes adhesion microbial surface components [66]. Similar putative pathways might be conserved in *Ostreobium*-associated *Ilumatobacter*, which might serve to adhere to algal host filaments. 

Additionally, amongst the core bacteria specific to each strain, the Phycisphaerales ASV322 SM1A02 lineage was detected only in thalli of 010. Phycisphaerales are known for their ability to break down and metabolize complex algal polysaccharides [67] that might be produced specifically by the 010 lineage. Phycisphaerales were also detected in the supernatants of strain 010, supporting the hypothesis of the bacterial utilization of potential algal-secreted polysaccharidic cell wall and mucilage components. 

### 4.2. Potential Intracellular Symbionts of Ostreobium

Intracellular bacteria are a still underexplored fraction of the coral holobiont and its associated Symbiodiniaceae and Ostreobineae (reviewed by Maire et al. [68]). Here, the existence of intracellular bacteria in *Ostreobium* thalli was revealed by direct FISH visualization of bacterial phylotypes within intact algal siphons of both strains, combined with persistent obtention of metabarcoded 16S rDNA sequences of two potential intracellular symbionts. 

*Candidatus ‘Amoebophilus’* lineage (ASV166; family Amoebophilaceae; order Cytophagales; class Bacteroidia) was ubiquitously detected at trace levels in all algal thalli of both studied strains. This lineage includes obligate intracellular bacteria with a reduced genome, discovered within a marine amoeba [69]. Recorded as a rare bacterial member of the coral holobiont [70], it was recently found at high abundances in the skeleton microbiome of several coral species (e.g., *Isopora prolifera* [10], *Pocillopora damicornis,* and Poritidae [54]), and we confirmed its detection within 75% of eight reef-sampled *Ostreobium*-colonized *Pocillopora* coral skeletons (see Appendix A). It was also very recently documented in a survey of the microbiota of a few different coral-isolated *Ostreobium* strains [27]. Here, its persistent detection in all 12 samples of each *Ostreobium* genotypes, at all three salinities, reveals a persistent association and suggests close and possibly intracellular association within *Ostreobium* siphons.

The Rickettsiales_AB1 lineage (ASV76) was also persistently detected in the thalli of both *Ostreobium* strains. Rickettsiales are obligate intracellular bacteria with reduced genomes that are known to infect algae; they were sequenced and visualized in the unicellular green alga *Carteria* (Volvocales) [71] and detected in two *Bryopsis* sp. species (Bryopsidales), sampled in the Pacific and North Sea at 18.0 and 32.4 psu [63]. Intracellular Rickettsiales have already been detected in coral host tissue (*Candidatus ‘Aquarickettsia rohweri’*; [72]) and in two Symbiodiniaceae microalgal strains, *Cladocopium goreaui* and *Durusdinium glynii*, which are isolated from the corals *Acropora tenuis* and *Porites lobata*, respectively [50]. The AB1 lineage highlighted here within cultured *Ostreobium* thalli was also detected in multiple Symbiodiniaceae cultures (propagated from an isolate from *Acropora tenuis* coral; [73]). 

### 4.3. Adjustments of Ostreobium Bacterial Community to Salinity Increase

At the ASV level, microbiota profiles were differentiated between low (32.9 psu) and high (40.2 psu) salinities in both algal genotypes, with intercalation of the intermediate salinity profile either with low (for strain 06) or high (for strain 010) salinity. Each strain (genotype) responded differently, with a seemingly higher salinity threshold for bacterial community changes in 06 than 010. At the order level (relative abundances barplot), proportions of Kiloniellales varied differently with salinity between strains, with an overall increasing trend with salinity in 06 (from cumulated abundances ~3 to ~7 to ~12%) vs. a decreasing trend in 010 (from cumulated abundances ~32% to 43% to 25%—except for core ASV976 and a few other ASVs). Proportions of Rhodospirillales also varied differently with salinity between strains, being lower at high salinity in strain 06 (from ~44% to ~25% cumulated abundances) vs. stable (at ~6.4 to 9.9% cumulated abundances) in strain 010. Finally, Nannocystales bacteria (Deltaproteobacteria) were detected only at high salinity and in strain 010 (ASV170).

Interestingly, both strains shared significantly increased proportions of Rhizobiales ASV597 (genus *Devosia*) at high salinity. In strain 010, the Hyphomonadaceae ASV57 (Caulobacterales) core bacteria were also significantly more abundant at high salinity. All together, the changes in the relative abundances of *Devosia* (Rhizobiales), Hyphomonadaceae (Caulobacterales), and Kiloniellales suggest the microbiota-mediated adjustment of the N metabolism in the algal holobiont at high salinity. Contrasted strategies between strains 06 and 010 are likely, supported by previous isotope tracer-based observations of different nitrate assimilation patterns [7]. Overall, the bacterial assemblages revealed here can be considered as extended phenotypes of each *Ostreobium* genetic lineage, supporting distinct algal metabolism, such as the distinct N and C assimilation and fatty acid metabolism previously recorded for 010 and 06 [7]. Core bacterial associates are likely to be selected by their capacity to use specific algal metabolites and secreted mucilage. Reciprocally, associated bacteria may contribute to limiting N, nutrients, and vitamins, sustaining algal host growth in the 33–41 psu salinity range. 

The microbiota salinity response observed here indicates flexibility that supports salt tolerance of this alga. Indeed, in other algal models, the composition and function of bacterial microbiota has an essential role in their host health and adaptation to salinity changes [74]. In the brown alga *Ectocarpus*, microbiota shifts in response to lower salinity were shown to be necessary for algal host survival [75]. In the cultured Ulvophyceae model *Ulva mutabilis*, the algal low salinity response involved interactions with marine bacteria *Roseovarius* and *Maribacter* [76]. The demonstrated flexibility of *Ostreobium* bacterial microbiota may be part of mechanisms that facilitate its adaptation to a large salinity range, supporting its ubiquitous distribution in carbonate reefs of contrasted salinities. This hypothesis is congruent with the recently proposed concept of microbiome flexibility contribution to accelerate acclimatization and adaptation of holobionts to environmental change [77]. 

### 4.4. Epiphytic vs. Endophytic Lifestyle of Ostreobium-Associated Bacteria

Indirect evidence for tight vs. loose bacterial associations can be derived from the observations of differential abundances of bacterial taxa between thalli and their supernatants. Here, a tight association is suggested for Rhodospirillaceae ASV16, Hyphomonadaceae ASV57, *Labrenzia* ASV72, *Candidatus ‘Amoebophilus’* ASV166, and *Ilumatobacter* ASV167 based on differential abundance patterns, which were 3- to 10-fold relatively more abundant in thalli than corresponding supernatants (carried over detection in supernatants likely via DNA amplification from potential residual *Ostreobium* fragments retained on filters). However, the nature of the Fodinicurvataceae association with *Ostreobium* still needs to be further investigated due to high sequence abundances in the supernatants of cultured strains, suggesting lifestyle traits of opportunistic bacteria that may be amplified by the in vitro conditions (nutrient-enriched culture medium). Finally, the potential intracellular Rickettsiales_AB1 lineage (ASV76) was equally or more abundant in supernatants than in *Ostreobium* thalli. This might be explained by parasitism on the dead portion of filaments (those without Dapi-stained nuclei) and/or utilization of N nutrients provided by decaying cells and culture medium. 

These examples illustrate the limitations to the tight/loose association hypotheses based on differential sequence diversity and abundances in thalli vs. supernatants. Additionally, due to the microscopic size and siphoneous cytological organization of this alga, microdissection attempts of individual filament content are highly likely to generate experimental cross-contaminations, further amplified by metabarcoding bias. Indeed, the cytology of siphoneous *Ostreobium* thalli, with cytoplasmic streaming in interwoven, branching, septae-lacking tubes [7,8], is a challenge for subsampling intact individual siphons for culture propagation and regionalized microbiota characterization. Thallus fragmentation involves a high risk of spilling the contents of broken siphons into the surrounding culture medium, which is a source of artifactual “supernatant” bacteria, as well as non-specific adhesion of released intracellular bacteria to the external sheath of broken filament as “epiphytic bacteria”.

Direct evidence of the intracellular/surface-associated/mucilage-associated localization of *Ostreobium* bacteria should thus be provided by in situ visualization of bacterial phylotypes at the scale of an intact tufts of filament to provide unambiguous arguments for endophytic vs. epiphytic lifestyles. 

Here, in situ hybridization (CARD-FISH) targeting 16S rRNA with universal bacterial probes allowed for the first visualization of metabolically active bacteria in or near *Ostreobium* siphons. Our study builds on previous FISH visualization of bacteria in the apical fragments of a related but freshwater *Bryopsis* alga [62]. However, in free-living *Ostreobium* growth forms (outside their carbonate habitat), morphological differentiation of spatial microniches at the algal thallus level was not possible. Indeed, orientation relative to a carbonate bioerosion front was lost. Orientation should be optimized so that bacteria can be precisely localized to specific siphon structures. For example, it has been shown in the macroscopically well-differentiated *Caulerpa* (Bryopsidale) thallus (fronds/rhizoids) that distinct epiphytic and endophytic bacterial communities existed [51,78] and were spatially differentiated from the rhizobiome [79]. Additionally, the relatively low spatial resolution of our epifluorescence study could be improved by confocal microscopy, in order to differentiate epiphytes from endophytes among bacterial phylotypes. Future CARD-FISH investigations will use taxon-specific probes designed from the sequences obtained in this study to visualize their distribution within *Ostreobium* thalli. Targeted taxa will include abundant Rhodospirillaceae and Kiloniellaceae, as well as rare but prevalent bacteria with putative intracellular lifestyles, such as *Candidatus Amoebophilus* and the Rickettsiales_AB1 lineage.

### 4.5. Partial Overlap of Bacteria Detected in Domesticated vs. Wild Ostreobium (within Corals)

Comparison of the communities of cultured algal strains with bacterial profiles from naturally occurring *Ostreobium*-colonized endolithic assemblages was used to evaluate the relevance of in vitro-based findings. Here, simplified in vitro inoculum (unialgal, long-term propagated *Ostreobium* cultures) and controlled stable environmental variables (only salinity was manipulated) revealed bacterial communities persistently associated to two specific *Ostreobium* genotypes. In field-collected *Ostreobium*-colonized coral carbonates, the endolithic bacterial community is shaped both by taxonomic composition of a large diversity of residents and by environmental variables, which fluctuate in time and between colonies and reef sites. This environmental complexity does not allow one to determine which bacterial fraction is specifically associated with which endolithic alga. Complexity is drastically reduced in culture-based “reductionist” experiments focusing on domesticated algal strains, compared with studies of native bacteria–*Ostreobium* assemblages within the carbonates of coral colonies.

Here, we show that the bacteria co-cultured with two distinct *Ostreobium* genotypes (strains) co-isolated from an aquarium-grown *Pocillopora* sp. coral colony [7] displayed partial diversity overlap with in hospite endolithic bacteria of environmental *Pocillopora* corals from Guam (Central Pacific) and Eilat (Northern Red Sea) (bacteria and *Ostreobium* diversity detailed in Appendix A). We confirmed the expected reduction in 16S rDNA ASV richness in “domesticated” *Ostreobium* unialgal strains, compared to “wild” *Ostreobium*-colonized *Pocillopora* coral endolithic assemblages from Guam and Eilat. Domesticated strains thus contained a fraction of co-isolated endolithic bacteria, able to survive in long-term aerobic cultures. This observation validates the use of *Ostreobium* strains as a simplified model for in vitro studies of the phycosphere of endolithic algae. 

Indeed, many (5) but not all seven bacterial taxa identified here as core *Ostreobium* microbiota were also found within the field collected corals, and/or have also been documented within skeleton assemblages of several coral species: Rhodospirillaceae ASV16 ([52], this study), Kiloniellaceae [54], Hyphomonadaceae ASV57 (this study), *Candidatus ‘Amoebophilus’* ([54], this study), and Rickettsiales_AB1 lineage ([72], this study).

*Labrenzia* (ASV72) Rhizobiales were not detected within coral skeletons, in contrast with their persistent detection within *Ostreobium* strains (this study). Building on their reported association with the Symbiodiniaceae coral tissue endosymbionts, this finding suggest that they may be associated to the algal partners and not the coral host. 

Future, a complementary investigation of core *Ostreobium* bacteria should involve bioeroding growth forms, complementary to the free-living growth forms studied here, to investigate a potentially different selection of bacterial associates in the more nutrient-limited and less oxygenated carbonate habitat, supporting *Ostreobium*’s changing metabolism across habitats [7].

Finally, bacterial community diversity is well-known to be biased by the selected DNA extraction method, molecular markers, and amplification primers. Here, the selected region V5–V7 of bacterial 16S rDNA gene limited co-amplification of algal plastidic 16S as shown in brown macroalgae [40,41] and *Caulerpa* [79]. Yet, amplification of the *tuf*A elongation factor, which is used for metabarcoding of *Ostreobium* algae ([5,80], this study), also yielded non-target bacterial *tuf*A sequences that were not detected via conventional 16S rDNA metabarcoding. This amplification of *tuf*A sequences from heterotroph bacteria in addition to phototroph cyanobacteria and *Ostreobium* algae was previously reported [5,80]. Our preliminary bacterial *tuf*A amplicon sequence analysis detected rare representatives of the Nitrospira phylum that remained undetected via V5–V7 16S rDNA metabarcoding: one ASV classified to Nitrospiraceae was detected per *Ostreobium* strain in thalli from 5/7 strains (strains 06, 017, 018B, 018C, and 019, not detected in 010 and 018A, described in Massé et al. [7]). Another rare alphaproteobacterial taxon revealed by *tuf*A sequencing was ASVs classified to the Xanthobacteraceae family, persistently detected at low abundance in 6/8 strains (1 ASV in 06, 1 ASV in 010, 2 ASVs in 017, 2 to 4 ASVs in 018A, and 1 ASV in 018B). Hence, the *Ostreobium*-associated bacterial diversity retrieved here cannot be considered exhaustive and is likely to be further extended using complementary markers.

## 5. Conclusions

In conclusion, this study of *Ostreobium* bacterial taxonomic diversity under experimental salinity stress shows that the microbiota of long-term-propagated strains was structured primarily by algal genotype, with salinity impacting the proportions of bacterial taxa putatively involved in N metabolism, between 32.9 and 40.2 psu. The identification of seven core bacterial taxa persistent across salinities and genotypes paves the way for further investigations of their roles in the physiology and overall health of their algal host. Bacteria-assisted metabolism may support *Ostreobium* metabolism and its adaptation to N-depleted environments and high salinity. More generally, the plasticity of microbiota composition highlighted here likely contributes to *Ostreobium*’s high ecological tolerance, supporting its ubiquitous distribution and resilient colonization of carbonates in changing reef salinities, and advocates for more in-depth approaches to their roles and importance in the coral holobiont.

## Figures and Tables

**Figure 1 microorganisms-11-01318-f001:**
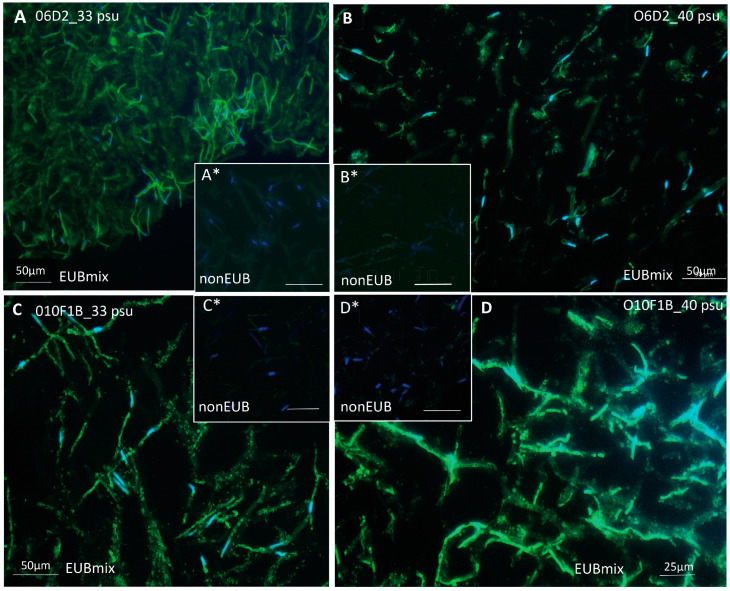
Visualization of bacteria in *Ostreobium* algal tissue sections via fluorescence in situ hybridization. Positive hybridization signal of universal bacterial probe mix (EUBmix) detected with Alexa488 (in green, cocci and rod-like morphotypes) on the surface and within *Ostreobium* filaments, or in the mucilage between filaments in deparaffinized sections (10 µm) of strains (**A**,**B**) 06 and (**C**,**D**) 010 cultured at (**A**,**C**) 32.9 psu and (**B**,**D**) 40.2 psu salinities. (**A***–**D***) Negative controls are hybridized with nonEUB probe. Algal nuclei counterstained with DAPI (in blue) are heterogeneously distributed within the coenocytic thalli, with multiple nuclei containing siphons alongside “empty” siphons (uneven Dapi staining in **A** and **D**). Alexa488-positive bacterial phylotypes are tightly associated to siphon envelopes (sheaths), with a few intra-siphon spots distinct from nuclei and chloroplasts. Positive Alexa488 signals also highlight cluster-aggregated bacterial phylotypes between algal filaments.

**Figure 2 microorganisms-11-01318-f002:**
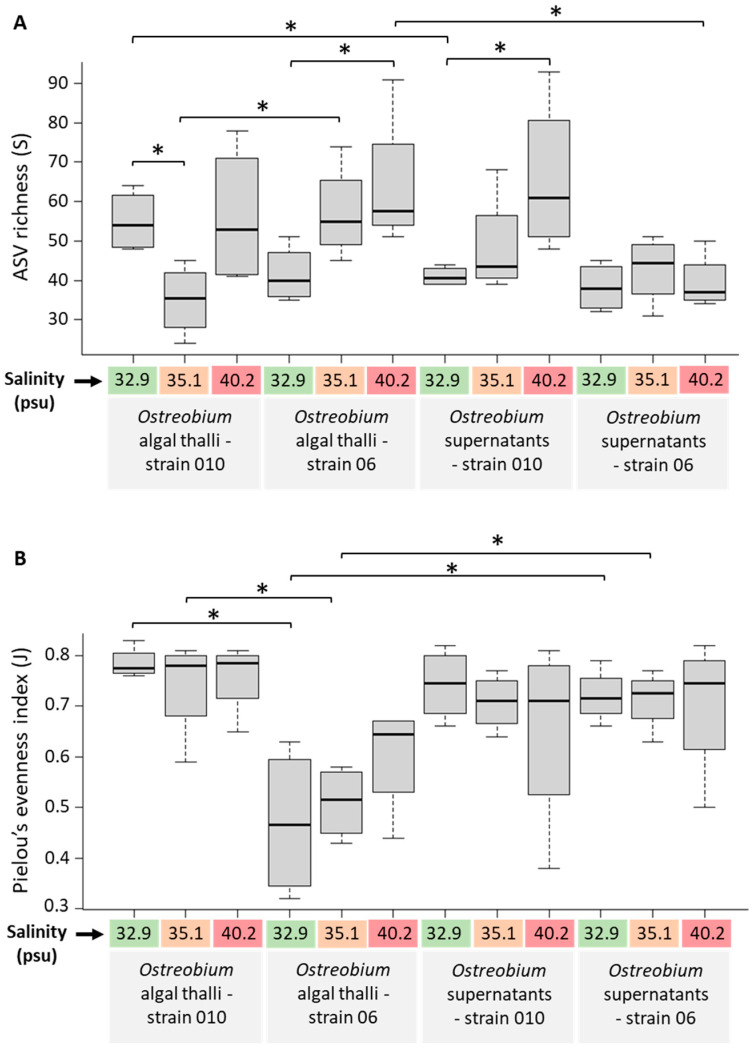
Alpha diversity of the bacterial communities (ASVs) of cultured *Ostreobium* thalli and their supernatants (depending on algal genotype and salinity). (**A**) ASV richness and (**B**) Pielou’s evenness index were calculated on the unrarefied ASV dataset. * Significant differences (Kruskal–Wallis, with pairwise post hoc Mann–Whitney test, *p* < 0.05) between comparisons of interest (e.g., between genotypes for the same salinity, between salinities for the same genotype, or between a thallus at a given salinity and its corresponding supernatant). Boxplots denote top quartile, median, and bottom quartile with *n* = 4 biological replicates/boxplot.

**Figure 3 microorganisms-11-01318-f003:**
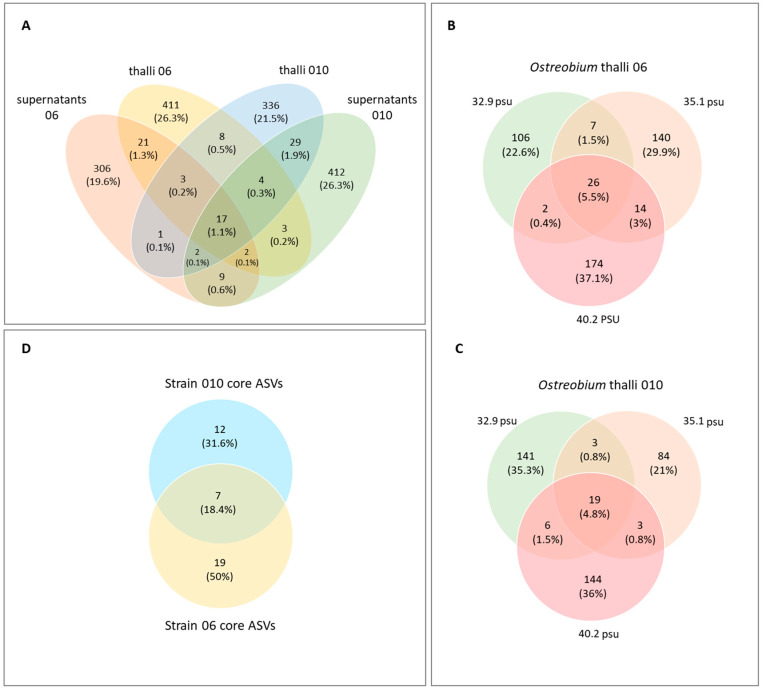
Venn diagrams of bacterial ASVs’ distribution (**A**) across cultured *Ostreobium* thalli and their corresponding supernatants; (**B**,**C**) across salinities (32.9, 35.1, and 40.2 psu) for *Ostreobium* thalli of strains (**B**) 06 and (**C**) 010. (**D**) Core microbiota of *Ostreobium* thalli shared by both algal genotypes: 010 *rbc*L clade P1 and 06 *rbc*L clade P14. Only ASVs conserved across 3 salinities for each genotype are represented (taxonomy and abundances detailed in Appendix A).

**Figure 4 microorganisms-11-01318-f004:**
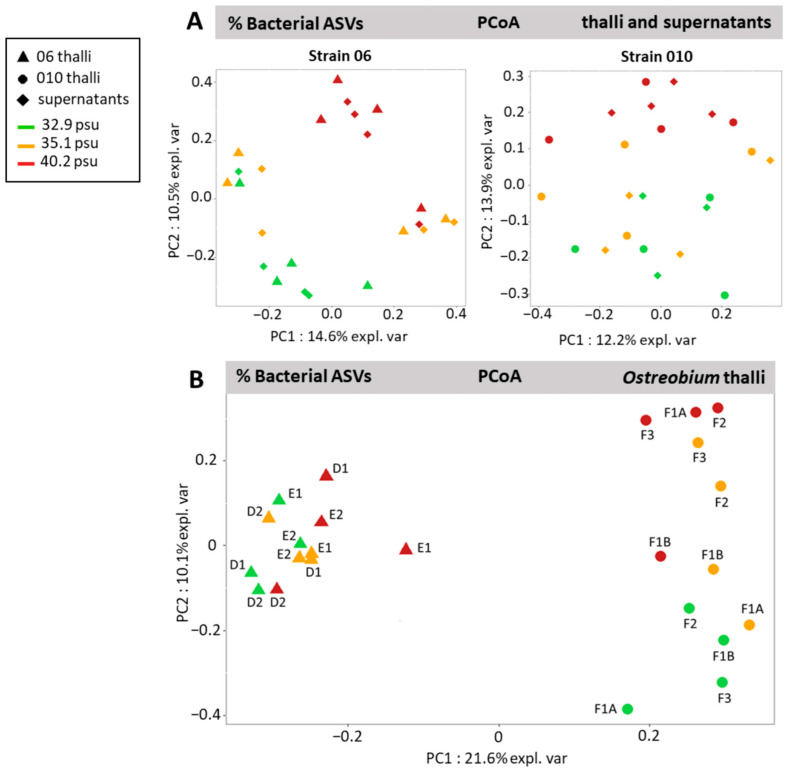
Structure of the bacterial communities visualized at the ASV level by Principal Coordinate Analysis of Bray–Curtis distances (arcsine square root-transformed proportions from unrarefied dataset). (**A**) Overlap between communities of cultured *Ostreobium* thalli and their corresponding supernatants for each genotype (strains 06 and 010) on Components 1 and 2. (**B**) Clustering by algal genetic lineage of *Ostreobium* thalli bacterial profiles (010: *rbc*L clade P1, 06: *rbc*L clade P14) with limited effect of salinity. Culture code names (FXX, EXX, DXX) are indicated in the score plot.

**Figure 5 microorganisms-11-01318-f005:**
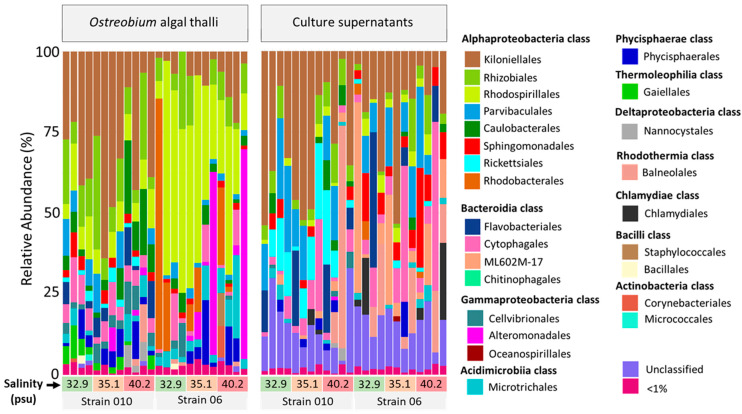
Taxonomic composition of bacterial orders in cultured *Ostreobium* thalli and their corresponding supernatants. Salinity and algal genotype are indicated below each individual sample column. The temperature was 25 °C. Orders detected at trace levels are aggregated in the <1% abundance category.

**Figure 6 microorganisms-11-01318-f006:**
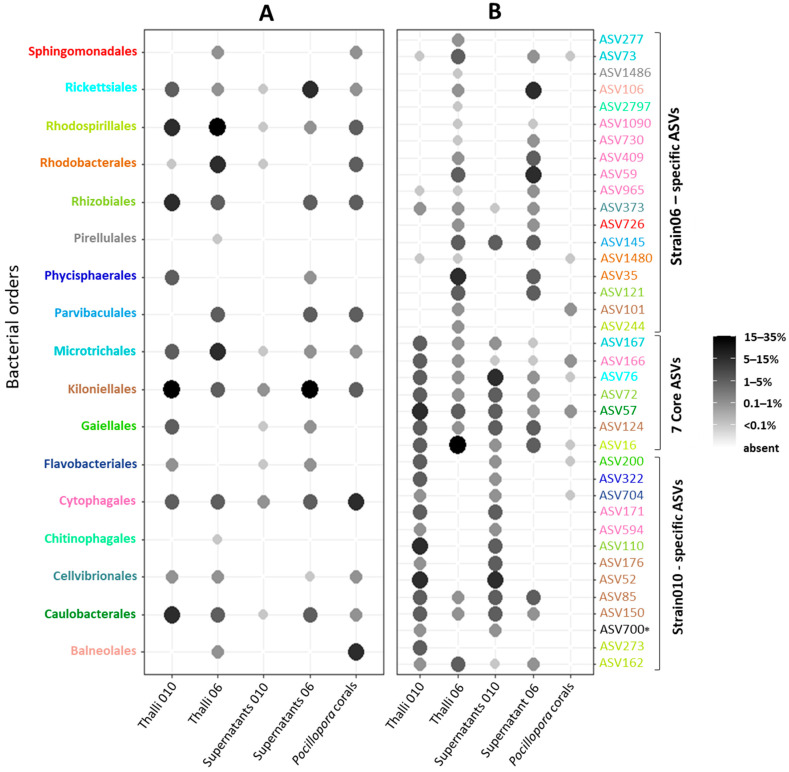
Core *Ostreobium* microbiota composition, shared at 3 salinities. The proportion of each (**A**) bacterial order or (**B**) ASV within a sample category is calculated as a percentage of total cumulated reads assigned to each category: thalli of strains 010 and 06, corresponding supernatants and environmental *Pocillopora* coral skeletons. Taxonomy classification after SILVA v138 SSU rRNA database released 16 December 2019. Abundances are grey-scale color and size-coded. Color code allows us to identify the (**B**) ASVs of each (**A**) bacterial order. * Bacterial order not determined.

**Figure 7 microorganisms-11-01318-f007:**
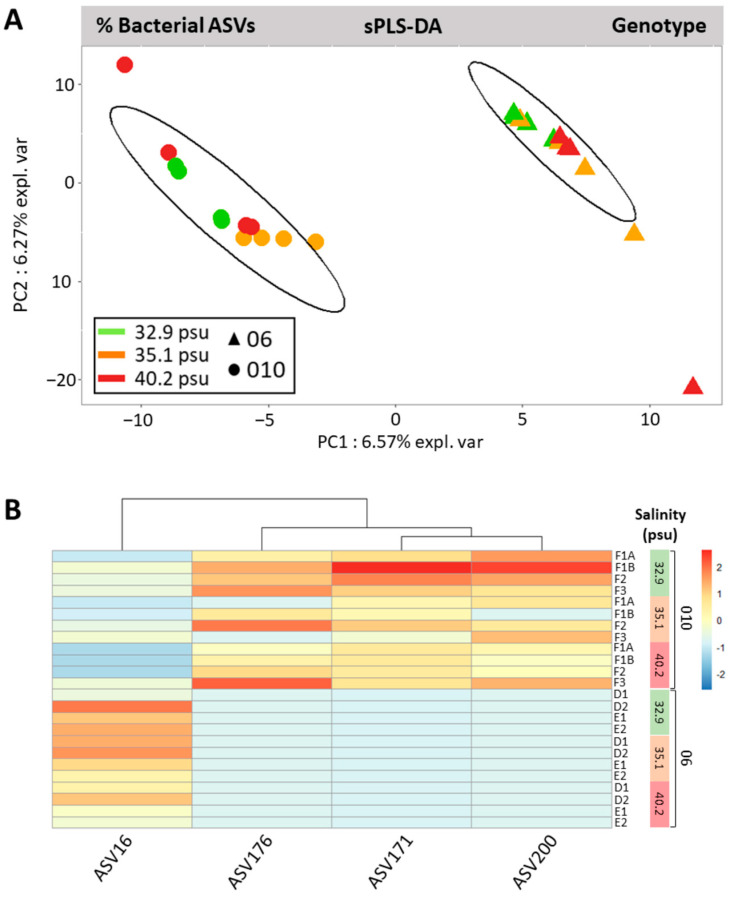
Algal genotype-driven structuration of bacterial ASVs communities in *Ostreobium* thalli. (**A**) Sparse Partial Least Squares Discriminant Analysis calculated from arcsine square root-transformed proportions of unrarefied dataset (sPLS-DA with 95% confidence statistical ellipses). (**B**) Heatmap of bacterial ASVs with correlation coefficient >0.7 on Component 1 of the PLS model, with color-coded variation of relative abundances (arcsine square root-transformed proportions; red indicates more abundant bacterial ASV, salinity levels for each strain 010 or 06 are reported on the right).

**Figure 8 microorganisms-11-01318-f008:**
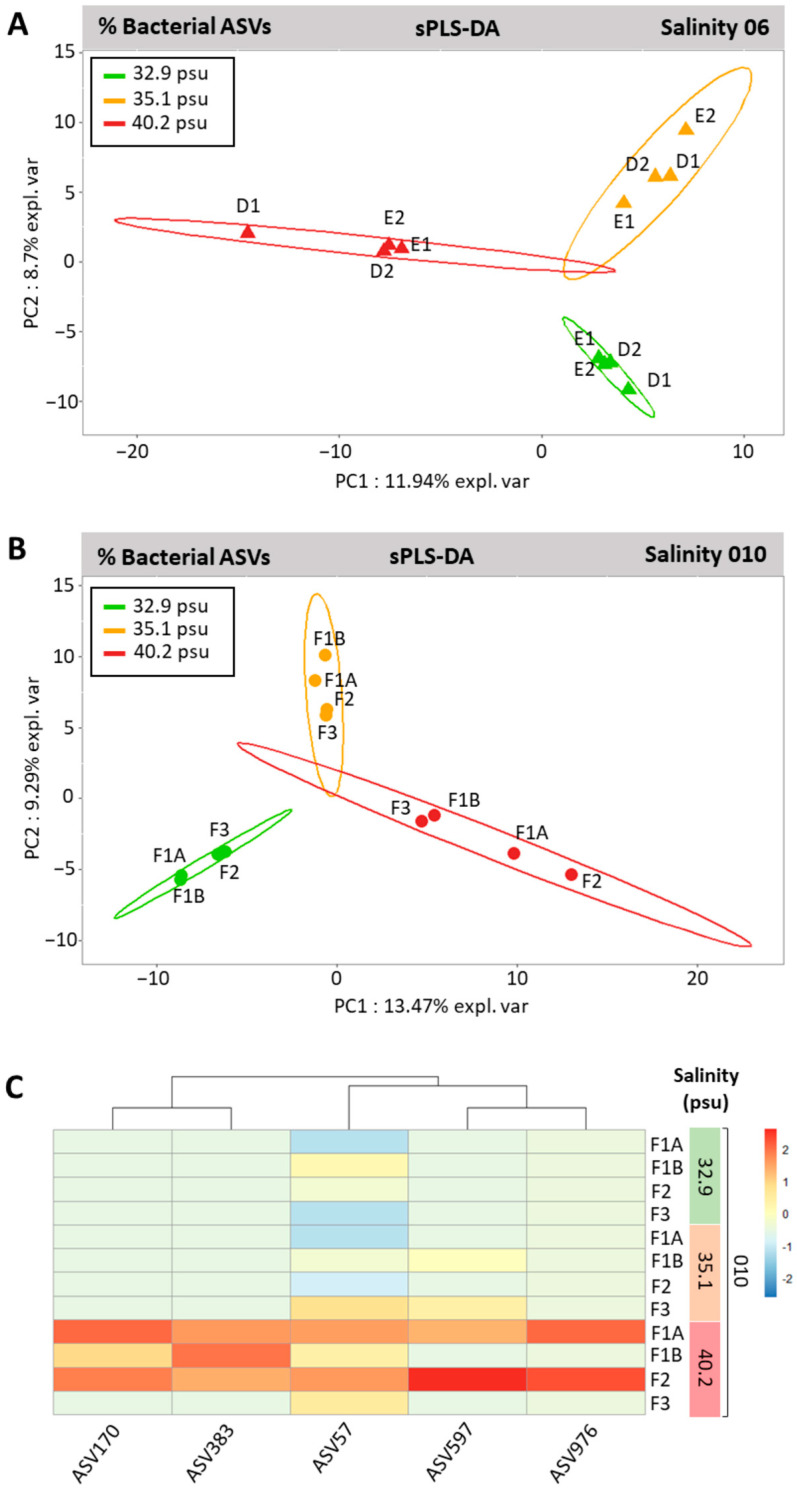
Salinity-responsive structuration of bacterial ASVs communities in *Ostreobium* thalli, within each genotype. (**A**) Strain 06 (*rbc*L clade P14) and (**B**,**C**) strain 010 (*rbc*L clade P1). (**A**,**B**) Sparse Partial Least Square Discriminant Analyses calculated from arcsine square root-transformed proportions of unrarefied dataset (sPLS-DA with 95% confidence statistical ellipses, and replicate culture code names (FXX, EXX, DXX)). (**C**) Heatmap of the 010-associated bacterial ASVs with correlation coefficient >0.7 on Component 1 of the PLS model, with color-coded variation of relative abundances (arcsine square root-transformed proportions; red indicates more abundant bacterial ASV). For 06, only 1 bacterial ASV with correlation coefficient >0.7 on Component 1 of the PLS model was identified (ASV597, enriched at 40.2 psu).

## Data Availability

Sequences generated via Illumina Miseq sequencing during this study were deposited as NCBI project archive under BioProject number PRJNA896951. Appendix A was deposited at figshare with doi: 10.6084/m9.figshare.21953075.

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
