# Peer review of "Bacterial Microbiota of Ostreobium, the Coral-Isolated Chlorophyte Ectosymbiont, at Contrasted Salinities"

_microorganisms, 2023, doi:10.3390/microorganisms11051318_

Round 1

Reviewer 1 Report

The authors present very detailed work on Ostreobium associated bacteria. This is interesting new avenue of research linked to coral microbiome. 

I have a several issue with the manuscript. I realize a significant amount of work was provided by the primary authors to give a comprehensive account of the study but some extremely lengthy part clutter the manuscript and burry the interesting parts in the details. I thus suggest the authors to drastically cut some parts of the manuscript (detailed below) and relay them to supplementary so that the work is not lost. In a revised manuscript, I hope that authors can remove the field samples as they clutter the manuscript and they futher focus on the difference between salinity and Ostreobium strains. Your manuscript could be a whole lot simpler that way. 

The Introduction is too long. When I get to line 94 I think it's the last paragraph because  paragraph starting line 76 seems to anounce the end of it...but then, there is again a full page of info. Please sort what is only truly necessary for the readers. Move other info somewhere else if needed. 

L94: There is another ongoing study on Ostreobium microbiome. Please see Pushpkumara et al in BioRXiv from the Verbruggen Lab "The bacterial microbiome of the coral skeleton algal symbiont Ostreobium shows preferential associations and signatures of phylosymbiosis".

Methods: OK

Results: I get lost in section 3.2. It's a bit difficult to relate strain# to families and the different experiences that were conducted. I would need a visual support tree/table or something similar to follow. I actually do not really see the point of having the Guam and Eilat samples in the study. They clutter the story which could much easier to tell without. You could keep those samples for another manuscript. 

Fig2AB. The Guam Eilat samples being from the field have crazy diversity compared to cultured Ostrebium. They squeeze down the richness of the cultured specimens and we don't see much. I really think the field samples are not necessary in this study and clutter your experimental/culture-based results. I would remove them and also remove the parentheses with the star since you do other multivariate analysis and significance tests later in the paper. Make a supplementary table with paairwise significance if you really wish to keep it.

Fig3. If remove field samples, then A would disappear. As is, I don't understand if the 8 ASVs that are common in A include the 7 ASVs that are common in B. In B, please add "Strain" in front of "06" and "010". Please do likewise in other figures to avoid confusion (e.g. Fig4).

Overall in Figure 3, it would be interesting to see Ostreobium thalli vs supernatant for the the two cultured strains separately. We have no visibility because you pool things directly in A.

Figure4. Likewise if field samples are removed it would be ok and authors could further focus the study on difference between thalli and supernatants per Ostreobium strain.

Section 3.4: There is way too much details for the reader to keep alert. Please make a summary paragraph of important findings (an overview) and move all this detail to the supplementary if people ever need to dig in the details they will go there.

Fig6. If I see correctly, the Pocillopora sample does not include the 7 common ASVs. Again why keep the field samples and clutter your findings and make the paper more complicated - focus on your experimental (culture) results. This figure can probably be improved. It makes us think that the diversity names on the left relate to ASV on the right. For B, you could position the common ASV in the middle, with strain 06 on top and strain 10 below. That would be more intuitive to see what's common in the middle.

Fig8 is interesting. Is Fig7 really important?

Discussion 4.1: it's normal that your field samples (a coral microbiome) are drastically richer that cultured Ostreobium. See my previous comments about inclusion of field samples in your study. This lengthen the manuscript, you could keep this for a future manuscript.

Discussion 4.2 is highly detailed again, too much info to follow. Provide necessary overview and move extra to supplementary.

Discussion 4.3: This should probably be the start of the discussion, that is what your manuscript is about. i.e. Results from cultures with salinity manipulations. 

Discussion 4.4: why not detail better the differences observed with salinity and Ostreo strain. The diversity may be similar, but the abundance of the different taxa seem to change a lot. 

Discussion 4.5: I cannot judge fully of CARD FISH results, please get the opinion of experts; however, the discussion does not seem to say that it was conclusive. It's a new approach and it is interesting to visualize the bacteria location (but since I am not an expert, even I know ostreobium, Fig1 is not easy to interpret or see what is to be seen). I realize this is quite difficult stuff to do technically. 

5 conclusion: Here again you seem to forget that you see diffference in abundance of taxa across salinities and should be an important part of your findings. 

Minor english:

Line 39, "as" move after exclusively

L42 "and". fix sentence, there is an issue. organelles and nuclei move by cytoplasmic streaming.

 I did not have time to check the supplementary content

Author Response

Manuscript ID microorganisms-2310844

Bacterial microbiota of Ostreobium, the coral-isolated chlorophyte ectosymbiont, at contrasted salinities.

Reviewer’s reports

Reviewer1

Comments and Suggestions for Authors

The authors present very detailed work on Ostreobium associated bacteria. This is interesting new avenue of research linked to coral microbiome. We thank Reviewer1 for this appreciation of our study.

I have a several issue with the manuscript. I realize a significant amount of work was provided by the primary authors to give a comprehensive account of the study but some extremely lengthy part clutter the manuscript and burry the interesting parts in the details. I thus suggest the authors to drastically cut some parts of the manuscript (detailed below) and relay them to supplementary so that the work is not lost. In a revised manuscript, I hope that authors can remove the field samples as they clutter the manuscript and they futher focus on the difference between salinity and Ostreobium strains. Your manuscript could be a whole lot simpler that way. We have followed Reviewer1’s constructive comments and focused the revised main text and figures (and abstract) on the algal cultures and their microbiota response to salinity. The data on field samples has been placed in Supplementary Material in order to contextualize the culture-based algal bacterial diversity relative to the complex, naturally occurring, endolithic diversity in coral skeletons. Comparison between environmental and cultured samples allows assessment of the bacterial fraction persistently associated to Ostreobium, i.e., co-isolated and long-term conserved across multiple algal subcultures, and hence of potential biological importance for the algal host.   

The Introduction is too long. When I get to line 94 I think it's the last paragraph because paragraph starting line 76 seems to announce the end of it...but then, there is again a full page of info. Please sort what is only truly necessary for the readers. Move other info somewhere else if needed. The revised Introduction has been shortened, with some info on coral endolithic diversity deleted. Information on salinity changes and their impact on reef corals and their microbiota (L86-104) have been kept to provide ecological context for the experimental manipulation presented in the study.     

L94: There is another ongoing study on Ostreobium microbiome. Please see Pushpkumara et al in BioRXiv from the Verbruggen Lab "The bacterial microbiome of the coral skeleton algal symbiont Ostreobium shows preferential associations and signatures of phylosymbiosis". We added this reference in the revised main text in the introduction (L84) and discussion (L656, L723) sections.

Methods: OK

Results: I get lost in section 3.2. It's a bit difficult to relate strain# to families and the different experiences that were conducted. I would need a visual support tree/table or something similar to follow. I actually do not really see the point of having the Guam and Eilat samples in the study. They clutter the story which could much easier to tell without. You could keep those samples for another manuscript. Following the Reviewer’s suggestion, we have removed this section and placed the detailed microbiota data of Guam and Eilat samples to the Supplementary Material. Reference to this environmental field material is kept in the Supplementary, in order to provide context for the culture-based experimental results. L502-506: “Additional information is provided (Figure 6, and details in Supplementary Material) on bacterial ASVs and orders also detected in environmental, naturally occurring Ostreobium-bacteria assemblages within coral skeletons, for contextualization of the core bacteria findings obtained in culture-based experiments.”

Fig2AB. The Guam Eilat samples being from the field have crazy diversity compared to cultured Ostreobium. They squeeze down the richness of the cultured specimens and we don't see much. I really think the field samples are not necessary in this study and clutter your experimental/culture-based results. I would remove them. We have revised Figure 2AB to remove the alpha diversity data of Guam and Eilat microbiota samples. These field-based results are now detailed in the Supplementary Material. and also remove the parentheses with the star since you do other multivariate analysis and significance tests later in the paper. Make a supplementary table with pairwise significance if you really wish to keep it. We kept in Figure 2A and 2B the parentheses with the star which indicate significant results (Kruskal-Wallis, with pairwise post-hoc Mann & Whitney test, p<0.05) since we think it is important to show these data directly on the figure and not in a separate table. Results of other, complementary significance tests are provided in Supplementary Table S4: “Table S4: Summary of pairwise permutational multivariate analysis of variance (‘adonis’ function) of Bray-Curtis distances (n=999 permutations, α = 0.05) for pairs of factor levels.”

Fig3. If remove field samples, then A would disappear. As is, I don't understand if the 8 ASVs that are common in A include the 7 ASVs that are common in B. In B, please add "Strain" in front of "06" and "010". Please do likewise in other figures to avoid confusion (e.g. Fig4). Overall in Figure 3, it would be interesting to see Ostreobium thalli vs supernatant for the two cultured strains separately. We have no visibility because you pool things directly in A. Following the reviewer’s suggestion, we have revised Figure3. Former Figure 3A has been moved to the Supplementary Material (to provide the overlapping bacterial fraction between cultured Ostreobium thalli and field endolithic bacteria). We added a new Venn diagram (revised figure 3A) to show the number of bacterial ASVs shared between Ostreobium thalli and supernatants for each strain but also to see the ASVs shared between thalli of each strain. We also clarified the text of the result section: “Between 9.2% and 13% of ASVs were shared between Ostreobium thalli and their corresponding supernatants for strain 06 and 010, respectively (Figure 3A).” (L382-385). We also added “Strain” in front of 06 and 010 on the revised Figure 3D (former Fig 3B).

Figure4. Likewise if field samples are removed it would be ok and authors could further focus the study on difference between thalli and supernatants per Ostreobium strain. Done. In revised Figure 4A, Principal Coordinate Analysis of Bray-Curtis distances now allows visualization of the variance of bacterial communities among thalli and supernatants for each Ostreobium strain.

Section 3.4: There is way too much details for the reader to keep alert. Please make a summary paragraph of important findings (an overview) and move all this detail to the supplementary if people ever need to dig in the details they will go there. This section has been shortened and is now focused on bacterial orders and ASVs found in the cultured thalli (with mention of their abundance in the corresponding supernatants). Contextual data on the detection of these bacterial orders and ASVs in field samples (coral skeletons) has been moved to the Supplementary.

Fig6. If I see correctly, the Pocillopora sample does not include the 7 common ASVs. The Figure 6 indicates that the Pocillopora samples contain 5 of the 7 Ostreobium core microbiota (common) ASVs. Again why keep the field samples and clutter your findings and make the paper more complicated - focus on your experimental (culture) results. In this figure 6, we decided to keep the field samples to identify shared ASV between cultured Ostreobium and environmental coral samples. Indeed, 5 of the 7 Ostreobium core microbiota ASVs (persistent across salinities and Ostreobium genotypes) were detected in skeletal bacterial fraction of field-samples (Ostreobium-colonized corals), validating the ecological significance of culture-based findings.  

This figure can probably be improved. It makes us think that the diversity names on the left relate to ASV on the right. For B, you could position the common ASV in the middle, with strain 06 on top and strain 10 below. That would be more intuitive to see what's common in the middle. Following the reviewer’s suggestion, we have moved the common ASV in the middle of the Figure 6B, with strain 06 on the top and strain 010 below. We also added a color code to identify the bacterial order affiliation (Figure 6A) of each core ASV (Figure 6B).

Fig8 is interesting. Is Fig7 really important? Figure 7 presents the results of statistical modeling (Partial Least Square regression) that discriminates between samples for the factor genotype. Compared to PCoA visualization of data variance structure, this approach adds statistical significance levels (95% confidence statistical ellipse) to the sample clustering observed by genotype.

Discussion 4.1: it's normal that your field samples (a coral microbiome) are drastically richer that cultured Ostreobium. See my previous comments about inclusion of field samples in your study. This lengthen the manuscript, you could keep this for a future manuscript. This section has been rewritten, shortened and combined with discussion on the comparison between culture-based and field samples “ 4.5. Partial overlap of bacteria detected in domesticated versus wild Ostreobium (within corals”) L829-843Comparison of the communities of cultured algal strains with bacterial profiles from naturally-occurring Ostreobium-colonized endolithic assemblages can be used to evaluate the relevance of in vitro-based findings. In culture-based experiments, simplified in vitro inoculum (unialgal, long-term propagated Ostreobium cultures) and controlled stable environmental variables (here only salinity was manipulated) revealed bacterial communities persistently associated to 2 specific Ostreobium genotypes. In field-collected Ostreobium-colonized coral carbonate samples, the endolithic bacterial community is shaped both by taxonomic composition of a large diversity of residents and by environmental variables, which fluctuate in time and between colonies and reef sites. This environmental complexity does not allow to determine which bacterial fraction is specifically associated with which endolithic alga. Complexity is drastically reduced in culture-based “reductionist” experimental approach focusing on domesticated algal strains, compared with studies of native bacteria-Ostreobium assemblages within marine carbonates including live coral colonies.”

Discussion 4.2 is highly detailed again, too much info to follow. Provide necessary overview and move extra to supplementary. This section has been shortened (“4.1. Small size core Ostreobium bacterial microbiota, structured by algal genotype” L628-705; 4.2. Potential intracellular symbionts of OstreobiumL707-736), summarizing the main findings in the context of previous studies.

Discussion 4.3: This should probably be the start of the discussion, that is what your manuscript is about. i.e. Results from cultures with salinity manipulations. We have extended this section to better discuss the results of the salinity response (“4.3. Adjustments of Ostreobium bacterial community to salinity increase”) L738-776.

Discussion 4.4: why not detail better the differences observed with salinity and Ostreo strain. The diversity may be similar, but the abundance of the different taxa seem to change a lot. We followed this constructive comment. L752-765 “Interestingly, both strains shared significantly increased proportions of Rhizobiales ASV597 (affiliated to the genus Devosia) at high salinity. In strain 010, the Hyphomonadaceae ASV57 (Caulobacterales) core bacteria was also significantly more abundant at highest salinity. All together the changes in Rhizobiales (Devosia), Hyphomonadaceae (Caulobacterales) and Kilonielalles relative abundances suggest microbiota-mediated adjustment of N metabolism in the algal holobiont at high salinity. Contrasted strategies between strains 06 and 010 are likely, supported by previous isotope tracer -based observations of different nitrate assimilation patterns [7]. Overall, the bacterial assemblages revealed here can be considered as extended phenotypes of each Ostreobium genetic lineage, supporting distinct algal metabolism, such as the distinct N and C assimilation and fatty acids metabolism previously recorded for 010 and 06 [7]. Core bacterial associates are likely to be selected by their capacity to use specific algal metabolites and secreted mucilage. Reciprocally, associated bacteria may contribute limiting N, nutrients, and vitamins, sustaining algal host growth in the [33-41] psu salinity range.”

Discussion 4.5: I cannot judge fully of CARD FISH results, please get the opinion of experts; however, the discussion does not seem to say that it was conclusive. It's a new approach and it is interesting to visualize the bacteria location (but since I am not an expert, even I know ostreobium, Fig1 is not easy to interpret or see what is to be seen). I realize this is quite difficult stuff to do technically.

The initial text insisted too much for a large readership on the technical limitations which are part of this approach but better discussed in a specialized microscopy journal. We have re-written the section to highlight the novelty of this method’s application to Ostreobium algae and have presented the limits (L328-330: “EUB-positive signal was also recorded within the filaments, revealing endophytic bacteria (with some uncertainty due to epifluorescence imaging corrected by signal deconvolution treatment).”) but also the benefits of this approach to reveal bacteria localization in situ, in intact algal filaments (L 808-814: “Direct evidence of the intracellular / surface-associated / mucilage associated localization of Ostreobium bacteria should thus be provided by in situ visualization of bacterial phylotypes at the scale of an intact tufts of filament, to provide unambiguous arguments for endophytic vs epiphytic lifestyles. Here, in situ hybridization (CARD-FISH) targeting 16S rRNA with universal bacterial probes allowed the first visualization of metabolically active bacteria in or near Ostreobium filaments.”)

Conclusion: Here again you seem to forget that you see difference in abundance of taxa across salinities and should be an important part of your findings. We modified the revised conclusion accordingly. L889-892: “In conclusion, this study of Ostreobium bacterial taxonomic diversity under experimental salinity stress shows that the microbiota of long-term-propagated strains was structured primarily by algal genotype, with salinity impacting the proportions of bacterial taxa putatively involved in N metabolism, between 32.9 and 40.2 psu. ”  

Minor english:

Line 39, "as" move after exclusively. Done

L42 "and". fix sentence, there is an issue. organelles and nuclei move by cytoplasmic streaming. We modified the sentence “with multiple nuclei and chloroplasts that move by cytoplasmic streaming” L41.

 I did not have time to check the supplementary content

Reviewer 2 Report

In general, the article is written and presented clearly and concisely, however, I have a few comments:
1. The main issue is the comparison:
The experiment reveals two types of algae grown in the laboratory at different salinity values. The comparison between these algae is clear.
However, comparing the algae to the coral skeletons and between the corals themselves is less relevant.
The corals were collected from two different areas (Eilat and Guam), which are exposed to different conditions, thus, the salinity is not the only factor.
Additional details, for example, when the corals were sampled (season), are missing. What was the water temperature? (even in the algae culture experiment, it is not mentioned, if I'm not mistaken).
I assume that If there were samples from the surrounding water of the coral skeleton samples, we would get a difference in the bacterial composition between the sites.
Therefore, it is inaccurate and even misleading to present that the difference in the microbiota between the corals results from the salinity alone (Figure 2 and Figure 5).
In addition, comparing algae grown for a long time in the laboratory (in a closed system and exposed to antibiotics (even if it was stopped before the experiment)) to coral skeletons in the sea, which are exposed to dynamic environmental conditions, is problematic. This variation is clearly visible in Figures 2-5, especially in Fig. 3A, which shows that only about 1% of ASVs overlap between the skeletons and algae. Therefore, comparing the alpha and beta diversity of the algae to the coral skeletons is not relevant, for example, in Fig. 2.

I understand that the goal was to find bacterial biomarkers for salinity that may also be found in skeletons in the field that are exposed to the same salinities. However, clarity and reservations are needed regarding the reason for the differences between the corals themselves and the algae experiment. And in particular, whether the changes in the microbiota population of the algae in the laboratory can reflect what is happening in the sea.
2. The experiment and results are pretty straightforward, but there seems to be some repetition of the Figures shown, for example, what is different regarding the message of Fig 4B to Fig 7A.
3. Please check the use of the terms rRNA/rDNA

Author Response

Manuscript ID microorganisms-2310844

Bacterial microbiota of Ostreobium, the coral-isolated chlorophyte ectosymbiont, at contrasted salinities.

Reviewer’s reports

Reviewer2

Comments and Suggestions for Authors

In general, the article is written and presented clearly and concisely, however, I have a few comments:

  1. The main issue is the comparison:

The experiment reveals two types of algae grown in the laboratory at different salinity values. The comparison between these algae is clear.

However, comparing the algae to the coral skeletons and between the corals themselves is less relevant. Comparison between the algae and the coral skeletons is provided to test the relevance of culture-based findings i.e.: the fraction of thalli associated bacteria also detected in naturally occurring Ostreobium-colonized coral endolithic assemblages, at similar salinities. For clarity and conciseness, the data on coral skeletons’ microbiota have now been moved to Supplementary text and figures. Comparisons between coral skeletons themselves are not detailed, because of the limited sampling effort. Here, as stated in the revised Supplementary section, comparison between environmental (field) and cultured (laboratory) samples allows assessment of the bacterial fraction persistently associated to Ostreobium, i.e., co-isolated and long-term conserved across multiple algal subcultures, and hence of potential biological importance for the algal host.    

The corals were collected from two different areas (Eilat and Guam), which are exposed to different conditions, thus, the salinity is not the only factor. We agree that the endolithic bacterial diversity of wild populations of corals is sensitive to many fluctuating environmental conditions, including salinity but also temperature and turbidity, in contrast to the controlled stable conditions of our laboratory-based experiments, where only salinity was manipulated. This remark has been added in the revised Main text, the Supplementary section, and the revised Supplementary Figure legends S4 and S5

Additional details, for example, when the corals were sampled (season), are missing. What was the water temperature? (even in the algae culture experiment, it is not mentioned, if I'm not mistaken). The water temperature has been added, as well as the sampling day (season), to the revised M&M and Supplementary Tables S2 and S3. The M&M indicates that the temperature for the algal culture experiment was incubator-controlled at 25°C, this information has now also been added to Figure S1 illustrating the culture experimental design. 

I assume that If there were samples from the surrounding water of the coral skeleton samples, we would get a difference in the bacterial composition between the sites. Therefore, it is inaccurate and even misleading to present that the difference in the microbiota between the corals results from the salinity alone (Figure 2 and Figure 5). We agree and for better clarity Figures 2, 3, 4 and 5 have been revised to remove the data relative to the coral skeletons. Data on the field collected samples is now placed in the Supplementary section with a sentence added to the legends of Figure S4 and S5: “Abiotic factors other than salinity fluctuate between coral field samples, affecting microbiota composition.”

In addition, comparing algae grown for a long time in the laboratory (in a closed system and exposed to antibiotics (even if it was stopped before the experiment)) to coral skeletons in the sea, which are exposed to dynamic environmental conditions, is problematic. This variation is clearly visible in Figures 2-5, especially in Fig. 3A, which shows that only about 1% of ASVs overlap between the skeletons and algae. Therefore, comparing the alpha and beta diversity of the algae to the coral skeletons is not relevant, for example, in Fig. 2. Figures 2, 3, 4 and 5 have been revised to remove the data relative to the coral skeletons, and the revised main text now focuses on the algal cultures at different salinities. We moved the Guam and Eilat coral skeleton microbiota data to the Supplementary Material (Supplementary text and Supplementary Figures). Results on this field-based material are detailed in order to provide context for the culture-based experimental results presented in the revised Main text. (Supplementary text paragraph: Comparison of bacterial associates from cultured Ostreobium to field occurring Ostreobium-bacteria assemblages; Supplementary Figures S3 and S4). In the main text, the revised discussion also states L844-854: “Here we show that the bacteria co-cultured with strains of 2 distinct Ostreobium genotypes co-isolated from an aquarium-grown Pocillopora sp. coral colony [7] displayed partial diversity overlap with in hospite endolithic bacteria of environmental Pocillopora corals from Guam (Central Pacific) and Eilat (Northern Red Sea) (environmental bacteria and Ostreobium diversity detailed in Supplementary Material). We confirmed the expected reduction in Amplicon Sequence Variant (16S rDNA ASV) richness in “domesticated” Ostreobium unialgal strains from culture collections, compared to “wild” Ostreobium-colonized Pocillopora coral endolithic assemblages from Guam and Eilat. Domesticated Ostreobium strains thus contained a fraction of co-isolated endolithic bacteria, able to survive in long-term aerobic cultures. This observation validates the use of Ostreobium strains as a simplified model for in vitro studies of the phycosphere of endolithic algae.”

I understand that the goal was to find bacterial biomarkers for salinity that may also be found in skeletons in the field that are exposed to the same salinities. However, clarity and reservations are needed regarding the reason for the differences between the corals themselves and the algae experiment. And in particular, whether the changes in the microbiota population of the algae in the laboratory can reflect what is happening in the sea. We agree and these differences are presented in the discussion section “ 4.5. Partial overlap of bacteria detected in domesticated versus wild Ostreobium (within corals”) L830-843Comparison of the communities of cultured algal strains with bacterial profiles from naturally-occurring Ostreobium-colonized endolithic assemblages can be used to evaluate the relevance of in vitro-based findings. In culture-based experiments, simplified in vitro inoculum (unialgal, long-term propagated Ostreobium cultures) and controlled stable environmental variables (here only salinity was manipulated) revealed bacterial communities persistently associated to 2 specific Ostreobium genotypes. In field-collected Ostreobium-colonized coral carbonate samples, the endolithic bacterial community is shaped both by taxonomic composition of a large diversity of residents and by environmental variables, which fluctuate in time and between colonies and reef sites. This environmental complexity does not allow to determine which bacterial fraction is specifically associated with which endolithic alga. Complexity is drastically reduced in culture-based “reductionist” experimental approach focusing on domesticated algal strains, compared with studies of native bacteria-Ostreobium assemblages within marine carbonates including live coral colonies.”

We also added the following sentence in the revised Supplementary section text: “The endolithic bacterial diversity of wild populations of corals is sensitive to fluctuating environmental conditions, including salinity but also temperature and turbidity, in contrast to the controlled stable conditions of laboratory-based experiments, where only salinity was manipulated.” The legends of revised Figure S4 and revised Figure S5 also state: “Abiotic factors other than salinity fluctuate between coral field samples, affecting microbiota composition.”

  1. The experiment and results are pretty straightforward, but there seems to be some repetition of the Figures shown, for example, what is different regarding the message of Fig 4B to Fig 7A. In Figure 4B, the variance of bacterial communities among algal genotypes and salinities is visualized by Principal Coordinate Analysis of Bray-Curtis distances. In Figure 7A a statistical model is further applied (Partial Least Square regression) to test the separation of communities according to factors algal genotype (Figure 7A) or salinity (Figure 8), with statistical ellipses (95% confidence) provided.
  2. Please check the use of the terms rRNA/Rdna. We have made corrections in the revised main text (L19, L116, L877).

Round 2

Reviewer 1 Report

The manuscript has been strongly shortened - thank you. Considering the format in which it is provided and the length of the manuscript, it is a bit difficult to see clear as a reviewer. A cleaned up version would have been more practical to review. Nonetheless, I did my best to make constructive comments and spot potential corrections

The results/discussion are still a bit long in my opinion and key results regarding important ASVs not easily seen from figures (specific ASVs discussed in the results/discussion). How to highlight them and showcase the most important ASVs discussed ?

Showing the abundance of core taxa is interesting (Fig6), but showing the abundance of the main taxa/ASV driving the separation of community in Fig4 (a bit like you did in Fig7C and Fig8C) may be more relevant to the results and discussion. This may mean making a heatmap or bargraph (or whatever format the author like) showing the abundance of ASVs explaining most of the variation on PC1 and PC2 (when thalli and supernantant are analysed together, see my comments further below about Fig4).

You may want to replace « filaments » by « siphons » throughout to be anatomically more correct or mention explicitly that « filaments » means « siphon » to you. In phycology, a filamentous algae is generally more descriptive of diminutive pluricellular algae.

I detail below a few comments and/or corrections to consider

Abstract

L16-17: Please review commas

L19: Prefer siphon to filament? that is also anatomically more correct

L22 Please add in parenthesis that host is Ostreobium strains species/lineage

L23-24 Both of these families are order Rhodospirales, it may be important to mention because it implies some degree of phylogenetic relatedness and importance of this order in siphonous algal evolution

L25 Please replace algal by ostreobium

L26 Either use only families or mention order in parenthesis for those we don’t know. I understand that some uncertainty in classification may push authors to use order or family. Try to be consistent.

Keyword suggestion :  endosymbionts, endophytes, endolithic, bioerosion ?

L47 You could start the sentence by « When endolithic in their bioeroding form, these filaments... »

L49 Please replace « its » by « substratum » to be more understandable

L69 See if you could move/merge with L49 since it is the same idea/topic

L86 By localization you mean intracellular or on the siphon surface?

L138 Perhaps add "in tropical locations adjacent to coral reef ecosystem", something like this

L178 and 179 It could be easier to name your strain OST06 and OST10 or a code according to your culture collection (for figures and text)

L191-193 The authors mention twice the antibiotics in the sentence. I believe you can rephrase to be more succinct and avoid the repetition while keeping all you want to say

L279 Do the author mean « average » rather than « each ». It does not seem correct to use each here

L313 Please correct to « chloroplast »

Fig2AB. Each boxplot lane is a cumulated richness of several replicates? e.g. strain 010 at 32.9 boxplot represent a pool of richness of multiple replicates? that is what I understand from the text. Is it correct to then test significance the way you did with this set up? please verify with a strong ecologist/statistician

Fig3. Do you explain prior what you mean by core ASVs? please check. Also if B and C could be on the same line (making D as B?), it may be easier to compare strain 06 and 010. Do you actually need D since you have Fig6 ? Personnally, when I read the paper and get to Fig3D I get confused...it’s only later in the paper that you refer to it.

Fig4. If you can, please remove the fig title in grey « % bacterial ASVs PCOA thalli and supernantant » in A and B. The caption explain or should explain it. It makes us think that each side of the title is equivalent to one of the plot below (left and write plots). Also why not show thalli + supernatants in B with corresponding colors and symbols, does it really clutter? Do the supernantant also separate when all points are in the same PCOA? You may even not need A then...?

Fig6. You kept Pocillopora corals there? Was it intended ? I am also not sure Fig6A is necessary.

Results 3.4 is still way too long for me. You have to find a way to show and explain only the most important aspects. There is too much details which looses the reader, I feel I am drowning in details and cannot find the take home message. Extra things can go in supplementary. Go back to your study goals and main conclusions to help you narrow down on the most important points.

I am not sure why authors opted for combined results discussion in the first place, that usually renders papers more difficult to swallow when there are lots of details. I am not asking to change it, it’s not my position to do so.

L932 please add: « in the microbiome as putative endophyte ». This study specifically sequenced blades/assimilator of Caulerpa as a simpler (less complex) microbiota than rhizoids which are probably extremely rich and would make putative endosymbionts difficult to spot. Other Caulerpa studies have sampled thalli in different places and used 16S which gave less clear results (amplification bias etc). Shotgun metagenomics probably gives more reliable profiles.

L1061. Any figure(s) supporting the statement to guide us readers?

L1212. add [5] to citation. With [80] these are the two primary studies that developped tools (tufA) to track Ostreobium via metabarcoding

L1216. Likewise, cite [80] here too

Author Response

Manuscript ID microorganisms-2310844 “Bacterial microbiota of Ostreobium, the coral-isolated chlorophyte ectosymbiont, at contrasted salinities”.

Reviewer’s reports

Reviewer1

The manuscript has been strongly shortened - thank you. Considering the format in which it is provided and the length of the manuscript, it is a bit difficult to see clear as a reviewer. A cleaned up version would have been more practical to review. Nonetheless, I did my best to make constructive comments and spot potential corrections. We thank the reviewer for the constructive effort and comments. Indeed, there was an error in the pdf version submitted (where track changes were visible) while the word version was a clean version (without track changes). Apologies presented.

The results/discussion are still a bit long in my opinion and key results regarding important ASVs not easily seen from figures (specific ASVs discussed in the results/discussion). How to highlight them and showcase the most important ASVs discussed?  The results/discussion section has been further reduced by 10 %. Heatmaps (Fig.7 & 8) and added color codes (Fig6) highlight the important ASVs driving microbiota differentiation among genotypes and salinities. We have strived to present each core ASV (descriptive list) discussed in the context of other algal microbiota studies. Reducing the text more carries the risk to not address precisely the complexity of this algal microbiota response to an abiotic stress factor (salinity). The journal does not impose size limits.

Showing the abundance of core taxa is interesting (Fig6), but showing the abundance of the main taxa/ASV driving the separation of community in Fig4 (a bit like you did in Fig7C and Fig8C) may be more relevant to the results and discussion. This may mean making a heatmap or bargraph (or whatever format the author like) showing the abundance of ASVs explaining most of the variation on PC1 and PC2 (when thalli and supernantant are analysed together, see my comments further below about Fig4). We understand and appreciate the constructive comment. But adding a heatmap in Fig4 would be redundant with the heatmap already in Fig 7B (L599-600 “(B) Heatmap of bacterial ASVs with correlation coefficient >0.7 on Component 1 of the PLS model, with color-coded variation of relative abundances”)

You may want to replace « filaments » by « siphons » throughout to be anatomically more correct or mention explicitly that « filaments » means « siphon » to you. In phycology, a filamentous algae is generally more descriptive of diminutive pluricellular algae. We agree. It has been replaced in the abstract and some sections of the text. But for better clarity to the non-specialist of the anatomy of these algae, we sometimes use both terms (in the abstract, first filament then L18 siphons). In the introduction the anatomy is defined using both terms: filament and siphon (L40). In the results section 3.1. In situ localization of bacteria associated to cultured Ostreobium siphons (L325-343) and in legend of Figure 1 (L345-355) both terms are used.

I detail below a few comments and/or corrections to consider

Abstract

L16-17: Please review commas. Done (2 commas removed)

L19: Prefer siphon to filament? that is also anatomically more correct. We agree. But for better clarity to the non-specialist of the anatomy of these algae, we use both terms in the abstract, first filament then siphons (L18).  In the introduction the anatomy is also defined using both terms: filament and siphon (L40).

L22 Please add in parenthesis that host is Ostreobium strains species/lineage. Done (L21)

L23-24 Both of these families are order Rhodospirales, it may be important to mention because it implies some degree of phylogenetic relatedness and importance of this order in siphonous algal evolution. Done, we mentioned that phylogenetic relatedness in parenthesis (L22). We also stressed that in the Discussion L629-631.

L25 Please replace algal by ostreobium. Done

L26 Either use only families or mention order in parenthesis for those we don’t know. I understand that some uncertainty in classification may push authors to use order or family. Try to be consistent. Done: we now use family-level classification for these taxa.

Keyword suggestion :  endosymbionts, endophytes, endolithic, bioerosion ? Done: we added the keyword endosymbionts (as the study focused on free-living growth form we opted not to use the keyword bioerosion).

L47 You could start the sentence by « When endolithic in their bioeroding form, these filaments... » Done, the sentence has been reformulated (L42-44).

L49 Please replace « its » by « substratum » to be more understandable. Done (L43).

L69 See if you could move/merge with L49 since it is the same idea/topic. We did not merge these sentences. Indeed, although the same topic, it is not the same idea, as L43 is the description of Ostreobium growth forms obtained in vitro (introducing the free-living growth form which is the focus of the MS), whereas L59-60 is the ecological context for those growth forms.

L86 By localization you mean intracellular or on the siphon surface? By localization we mean the position of the bacteria relative to the siphon. This has been clarified (L77: “Visualization of bacteria in intact algal siphons (intracellular or at surface) is an open challenge”)

L138 Perhaps add "in tropical locations adjacent to coral reef ecosystem", something like this. Done (L96).

L178 and 179 It could be easier to name your strain OST06 and OST10 or a code according to your culture collection (for figures and text). We appreciate this suggestion but prefer to use the code names already published for Ostreobium strain 010 and Ostreobium strain 06 in Massé et al. 2020 Envir. Microbiol, and deposited in the MNHN culture collection.

L191-193 The authors mention twice the antibiotics in the sentence. I believe you can rephrase to be more succinct and avoid the repetition while keeping all you want to say. We have modified the sentence to avoid this repetition: L137-140 « Ostreobium strains were originally isolated from their coral host and propagated for 4.5 years with low doses of penicillin (100 U/ml) and streptomycin (100 µg/mL) antibiotics to prevent overgrowth by opportunistic bacteria [7]. »

L279 Do the author mean « average » rather than « each ». It does not seem correct to use each here. OK, the term ‘each’ was removed.

L313 Please correct to « chloroplast ». Done

Fig2AB. Each boxplot lane is a cumulated richness of several replicates? e.g. strain 010 at 32.9 boxplot represent a pool of richness of multiple replicates? that is what I understand from the text. Each boxplot illustrates the data obtained from 4 biological replicates, from which median and quartiles are inferred (see revision caption, in which this was added). We added this information in the legend « n=4 biological replicates/boxplot » L394. Is it correct to then test significance the way you did with this set up? please verify with a strong ecologist/statistician. Since 4 replicates do not allow parametric tests, we used a non-parametric Kruskal-Wallis test followed by Mann&Whitney post-hoc test, which is a standard procedure for non-parametric data distribution in microbial ecology datasets.

Fig3. Do you explain prior what you mean by core ASVs? please check. Also if B and C could be on the same line (making D as B?), it may be easier to compare strain 06 and 010. Do you actually need D since you have Fig6 ? Personnally, when I read the paper and get to Fig3D I get confused...it’s only later in the paper that you refer to it. The definition of the core microbiota is in the legend of Fig3D (L411-413) “Core microbiota of Ostreobium thalli shared by both algal genotypes: 010 rbcL clade P1, 06 rbcL clade P14. Only ASVs conserved across 3 salinities for each genotype are represented”). We have now also added this definition L375 “(between strains comparison of thalli (Figure 3D) define the core microbiota, see 3.4)”. For easier comparison of strains 06 and 010 we have modified Figure 3 to put in the same column Fig 3B (strain 06) and Fig 3C (strain 010). We believe it is important to keep Fig3D for synthetic illustration of the core ASVs, further detailed in Figure 6.

Fig4. If you can, please remove the fig title in grey « % bacterial ASVs PCOA thalli and supernantant » in A and B. The caption explain or should explain it. It makes us think that each side of the title is equivalent to one of the plot below (left and write plots). We have clarified that the title “% bacterial ASVs PCoA thalli and supernantant” applies to both graphs below by removing the highlight for the mentions “strain 06” and “strain 010”. It is in the caption indeed, but we wish to keep this information visible in the graph to easily differentiate at first glance the data visualization method in Fig 4 (PCoA projection) from the data statistical modeling method in Fig 7A (Partial Least Square regression, with statistical ellipses reported on the graph)   

Also why not show thalli + supernatants in B with corresponding colors and symbols, does it really clutter? Do the supernantant also separate when all points are in the same PCOA? You may even not need A then...? As we have 3 variables (salinity, genotype, nature i.e. thalli/supernatant) putting all the information into one single graph really clutters the information (poor readability). Moreover, a previous request was to see clearly the data distribution between thalli and supernatant for the two cultured strains separately.

Fig6. You kept Pocillopora corals there? Was it intended? In Figure 6, we decided to keep the field samples to identify shared ASV between cultured Ostreobium and environmental coral endolithic samples, for ecological contextualization. Indeed, 5 of the 7 Ostreobium core microbiota ASVs (persistent across salinities and Ostreobium genotypes) were also detected in skeletal bacterial fraction of field-samples (Ostreobium-colonized corals), validating the ecological significance of culture-based findings.  I am also not sure Fig6A is necessary. To our opinion, Figure 6A allows synthetic illustration of the core microbiota at higher (order) taxonomical level, complementary to ASV level. This Figure thus summarizes the text of section 3.4. We also added color codes to identify the bacterial order affiliation of each core ASV, as requested previously.

Results 3.4 is still way too long for me. You have to find a way to show and explain only the most important aspects. There is too much details which looses the reader, I feel I am drowning in details and cannot find the take home message. Extra things can go in supplementary. Go back to your study goals and main conclusions to help you narrow down on the most important points. This section Results 3.4 has been further reduced by 16 %, keeping the most important findings necessary to support the discussion (comparison of microbiota across Bryopsidales algae, …). We believe that a list is unfortunately unavoidable to present each ASV of the core microbiota, and explain the take home message of Figure 6 and Supplementary materials.

I am not sure why authors opted for combined results discussion in the first place, that usually renders papers more difficult to swallow when there are lots of details. I am not asking to change it, it’s not my position to do so.

L932 please add: « in the microbiome as putative endophyte ». Done

This study specifically sequenced blades/assimilator of Caulerpa as a simpler (less complex) microbiota than rhizoids which are probably extremely rich and would make putative endosymbionts difficult to spot. Other Caulerpa studies have sampled thalli in different places and used 16S which gave less clear results (amplification bias etc). Shotgun metagenomics probably gives more reliable profiles.

L1061. Any figure(s) supporting the statement to guide us readers? This statement is supported by the barplot graph of bacterial orders relative abundances (Fig 5 & section 3.4) now mentioned in parenthesis (L723)

L1212. add [5] to citation. With [80] these are the two primary studies that developped tools (tufA) to track Ostreobium via metabarcoding. Done

L1216. Likewise, cite [80] here too. Done
